# C/EBPB-dependent adaptation to palmitic acid promotes tumor formation in hormone receptor negative breast cancer

Xiao-Zheng Liu [1,7], Anastasiia Rulina[1,7], Man Hung Choi[2,3,7], Line Pedersen[1], Johanna Lepland[1], Sina T. Takle [1], Noelly Madeleine[1], Stacey D'mello Peters [1], Cara Ellen Wogsland [1], Sturla Magnus Grøndal [1], James B. Lorens[1], Hani Goodarzi[4], Per E. Lønning[5,6], Stian Knappskog [5,6], Anders Molven [2,3] & Nils Halberg [1✉]

Epidemiological studies have established a positive association between obesity and the incidence of postmenopausal breast cancer. Moreover, it is known that obesity promotes stem cell-like properties of breast cancer cells. However, the cancer cell-autonomous mechanisms underlying this correlation are not well defined. Here we demonstrate that obesity-associated tumor formation is driven by cellular adaptation rather than expansion of pre-existing clones within the cancer cell population. While there is no correlation with specific mutations, cellular adaptation to obesity is governed by palmitic acid (PA) and leads to enhanced tumor formation capacity of breast cancer cells. This process is governed epigenetically through increased chromatin occupancy of the transcription factor CCAAT/enhancer-binding protein beta (C/EBPB). Obesity-induced epigenetic activation of C/EBPB regulates cancer stem-like properties by modulating the expression of key downstream regulators including *CLDN1* and *LCN2*. Collectively, our findings demonstrate that obesity drives cellular adaptation to PA drives tumor initiation in the obese setting through activation of a C/EBPB dependent transcriptional network.

[1] Department of Biomedicine, University of Bergen, N-5020 Bergen, Norway. [2] Gade Laboratory for Pathology, Department of Clinical Medicine, University of Bergen, N-5020 Bergen, Norway. [3] Department of Pathology, Haukeland University Hospital, N-5021 Bergen, Norway. [4] Department of Biophysics and Biochemistry, University of California San Francisco, San Francisco, CA 94158, USA. [5] Department of Clinical Science, Faculty of Medicine, University of Bergen, N-5020 Bergen, Norway. [6] Department of Oncology, Haukeland University Hospital, N-5021 Bergen, Norway. [7]These authors contributed equally: Xiao-Zheng Liu, Anastasiia Rulina, Man Hung Choi. ✉email: nils.halberg@uib.no

Breast cancer is the most frequently diagnosed cancer and a leading cause of cancer-related death among women. Risk factors for breast cancer consist of non-modifiable factors, such as age, genetic predisposition, and reproductive history, and modifiable factors such as obesity and alcohol consumption and tobacco smoking. As an independent risk factor, postmenopausal (PM) obesity accounts for up to 20% higher risk of developing breast cancer, and every 5-unit increase in BMI is associated with a 12% increase in breast cancer risk[1]. Whereas obesity in PM individuals has been consistently linked to higher risk of developing estrogen receptor (ER) positive breast cancer, the effect in ER-negative breast cancer has been more debated[2]. In addition to effects on breast cancer risk, meta-analyses have suggested that overweight and obesity are associated with worse overall survival and metastasis-free survival independent of menopause or hormone receptor status[3–5].

Work in mouse models generally recapitulates both obesity-linked tumor initiation and progression[6]. Of these, tumor progression has been most extensively studied and proposed mechanisms include obesity-induced chronic inflammation[7,8], altered insulin signaling[9], deregulation of estrogen[10], rewiring of cancer metabolism[11], and adipokine secretion[12]. Recent insights into obesity-dependent regulation of tumor initiation in breast cancer have highlighted a number of non-cell-autonomous mechanisms including regulation of metabolically activated macrophages[13], leptin[14], and FABP4[12].

Here, we aimed to identify cancer cell-autonomous determinants of obesity-induced PM breast cancer risk. We demonstrate that obesity has adverse effects on patient survival in PM, ER/progesterone receptor (PR) negative breast cancers compared to other subtypes. We show that cellular adaptation to obese environments is phenotypically and mechanistically recapitulated by long-term exposure to high concentrations of palmitic acid (PA) in vitro. Both obesity and long-term adaptation to high levels of PA engender cancer cell dedifferentiation towards stem cell-like properties in both human biobank material and mouse models. Mechanistically, we identify epigenetic activation of the CCAAT/enhancer-binding protein beta (C/EBPB) transcription factor as a required regulator of obesity-induced cancer stem-like properties. We further demonstrate that C/EBPB governs cancer stemness through the modulation of CLDN1 and LCN2. Taken together, our findings demonstrate that cellular adaptation to obesity-induced PA is a key driver of tumor initiation in PM/ER−/PR− breast cancer cells in obesity.

## Results

**Obesity is associated with increased frequency of stem cell-like cancer cells in PM/ER−/PR− breast cancer patients and mouse models of breast cancer.** To quantitively determine how obesity is linked to increased risk of breast cancer, we orthotopically implanted E0771 and TeLi (Wnt1-driven) cells at limiting dilutions in a C57BL/6J diet-induced obesity model and measured tumor formation. High-fat diet (HFD) feeding resulted in weight gain, and HFD-fed mice displayed multiple hallmarks of obesity-induced comorbidities such as liver steatosis, hyperinsulinemia, hyperglycemia, and reduced glucose clearance compared to the regular chow diet-fed mice (Fig. S1A–D). Following mammary fat pad implantation of limiting numbers of E0771 and TeLi cells, we demonstrate that high-fat environments consistently promote tumor formation with a 6–10-fold enrichment in cancer stem cell frequencies (Fig. 1a). This is consistent with a previous report[13] that focused on non-cancer cell-autonomous regulation linking obesity to breast cancer initiation. In contrast, we set out to investigate how the distorted metabolic environment affects cancer cell behavior and thereby gain insights into potential cancer cell-autonomous mechanisms that drive breast cancer initiation in obese environments. To establish the framework for such mechanistic studies, we first sought to identify a group of patients affected by the obese state. To that end, we performed survival analyses of 115 PM (defined by age of >50 years) breast cancer patients using BMI and hormone (estrogen and progesterone) receptor status as variables in a highly controlled in-house dataset[15]. Overweight and obesity (BMI > 25) were associated with significantly reduced disease-specific survival rates in hormone receptor-negative patients as compared to non-obese patients (Fig. 1b). No effects of BMI on disease-specific survival were observed in the hormone receptor-positive patients despite having equal BMI distribution as hormone receptor-negative patients (Fig. S1E, F). Importantly, within the PM/ER−/PR− patient group there were no differences between the high and low BMI groups with respect to patient age (Fig. S1G), tumor size (Fig. S1H), or tumor stage (all included patients were stage 3) at the time of diagnosis.

To examine potential cancer cell-specific adaptations in the in vivo tumor microenvironment of obese and non-obese patients, we obtained tumor tissue microarrays (TMA) from this group of PM/ER−/PR− patients and immunostained the cores for the stemness markers CD133[16,17] and Axl[18]. The image analysis platform QuPath[19] was used to segment the images, differentiate between stromal and cancer cells and to quantify cancer cell-specific CD133+ and Axl^high cell frequencies. PM/ER−/PR− breast cancer patients with a BMI above 25 displayed both higher CD133+ and Axl^high cancer cell frequencies as compared to the normal BMI patients (Fig. 1c, d). This suggests that adaptation to obese environment leads to an enrichment in cancer stemness in both mice models and breast cancer patients.

To investigate how the obese environment affects cancer cells phenotypes, we hypothesized that the cellular adaptations induced by obesity are maintained ex vivo. We therefore dissociated tumors formed in the obesogenic and non-obesogenic environments and sought to identify deregulated cellular traits. While cellular proliferation was unaffected (Fig. 1e), the ability to form tumorspheres was significantly enhanced following adaptation to the obesogenic environment (Fig. 1f). After isolation from tumors formed in obese mice, ex vivo E0771 cells displayed metabolic rewiring rendering the cells more reliant on PA oxidation and less reliant on glucose oxidation (Fig. 1g). Both enhanced tumorsphere formation capacity[20–22] and metabolic reprogramming[23,24] has been linked to stem cell behavior in breast cancer and are thus consistent with cancer cell-autonomous effects of obesity.

**Long-term adaptation to PA phenocopy obesity-induced stem-cell features.** We next wondered if the obese environment selects for a pre-existing clone or governs generalized adaptation. To this end, we tagged individual E0771 cells with a high complexity DNA barcode library (ClonTracer)[25] to track the fate of individual cancer cells as tumors formed in obese and non-obese environments. Generally, we observed a high variability of unique barcodes in tumors evolving in lean as well as in obese mice. The three replicates presented different abundances of barcodes, and the enrichment of certain subset of barcodes was not presented in all replicates (Fig. S2A). We then compared the barcode distribution of relative barcode size between the tumors derived from chow and HFD-fed mice and demonstrate that the overall barcode distributions were unaffected by the different diets, suggesting that exposure to an obese environment did not select for a pre-existing clone within the subpopulations (Fig. 2a).

Obesity leads to the production of reactive oxygen species in adipose tissue[26]. Given the abundant adipose tissue in the

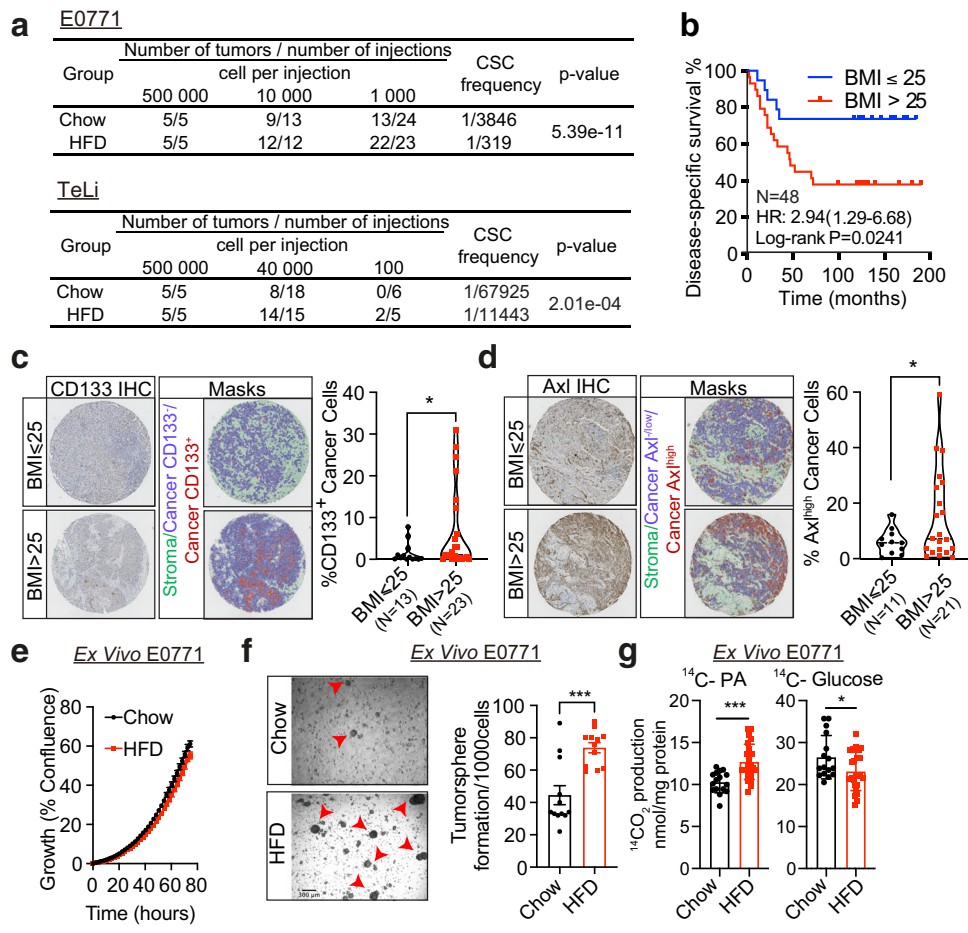

**Fig. 1 Obesity is associated with increased frequency of stem cell-like cancer cells in PM/ER⁻/PR⁻ breast cancer patients and mouse models of breast cancer. a** Tumor incidence following orthotopic implantation of cancer cells into the mammary fat pads of chow- and HFD-fed mice. **b** Kaplan–Meier curves show disease-specific survival for PM/ER⁻/PR⁻ patients ($n = 48$) with high (red) or low (blue) BMI. Log-rank (Mantel–Cox) $P$ value is denoted for difference in disease-specific survival. **c**, **d** Representative tissue microarray and QuPath analysis mask pictures of CD133 (**c**) and Axl (**d**) staining in high (BMI > 25, $n = 23$ for CD133; $n = 21$ for Axl) or low (BMI ≤ 25, $n = 13$ for CD133; $n = 11$ for Axl) BMI PM/ER⁻/PR⁻ patients' tumor samples, stroma is marked in green, CD133⁻ or Axl⁻/low cancer cells are marked in blue and positive staining cancer cells are marked in red (for **c**, $P = 0.0245$; **d**, $P = 0.023$). **e** Time-dependent proliferation assay of ex vivo E0771 cells isolated from chow or HFD-fed mice. For each time point, data are represented as mean ± SEM of four tumor samples (eight replicates per tumor sample were measured) from each group. **f** Tumorsphere formation assay of E0771 ex vivo cells isolated from chow diet or HFD-fed mice. Representative images of day-5 tumorspheres formation of E0771 ex vivo cells and red arrowheads mark the identified tumorspheres. Quantification of day-5 tumorspheres is represented as mean ± SEM of four tumor samples from each group and three replicates were measured for each sample, $P = 0.0002$. **g** Fatty acid and glucose oxidation on E0771 ex vivo cells isolated from two chow diet-fed mice and three HFD-fed mice. The oxidation data are normalized to cell protein content. For each sample, 8 replicates are measured and data are represented as mean ± SEM of all replicates from each group. Fatty acid oxidation $P = 0.0002$, Glucose oxidation $P = 0.0365$. For **c**, **d**, unpaired, two-tailed Welch's $t$-test was used for statistical testing. For **f**, **g**, statistical significance determined with unpaired, two-tailed Student's $t$-test (*$P$ value <0.05; ***$P$ value <0.001). Source data are provided as a Source data file.

mammary gland and the association between reactive oxygen species and mutagenesis[27], we then asked if the obese phenotype enriches for specific mutations. We therefore performed high-coverage sequencing of 360 known cancer genes[28] in tumor samples collected from the PM/ER⁻/PR⁻ patient group at the time of diagnosis. Based on the analysis we were not able to detect any mutations correlating to obesity across this panel (Supplementary Table S1). The combined results from the in vivo barcode studies and patient analysis suggest that obesity governs cellular adaptation independent of obesity-dependent genetic changes.

To understand what drives such cellular adaptation we reasoned that PA could play a central role as (i) the circulating abundance of PA strongly elevated in obese individuals reaching levels that are toxic to cancer cells and, thus, could feasibly provide a new selection pressure[29–31], (ii) PA has been reported

to be epidemiologically associated with a higher risk of developing PM breast cancer[32,33], and (iii) we (Fig. 1) and others demonstrate that cancer cell PA metabolism is altered by obesity[7]. To assess how breast cancer cells adapt to PA exposure, we cultured hormone receptor-negative breast cancer cell lines to increasing PA concentrations over a period of 2 months. Human (MDA-MB-231 and HCC1806) and mouse (E0771) breast cancer cells consistently adapted to acquire resistance to PA-induced apoptosis to enable persistent growth even in the high PA environment (Fig. 2b; Fig. S2B). For adapted MDA-MB-231 (MDAapa) and HCC1806 (HCC1806apa), acquired resistance was accompanied by a reduction in growth rate, whereas adapted E0771 (E0771apa) cells maintained its growth rate even after adaptation to high levels of PA (Fig. 2c; Fig. S2C). Importantly, the final PA concentration is similar to the serum levels of PA in obese individuals[29].

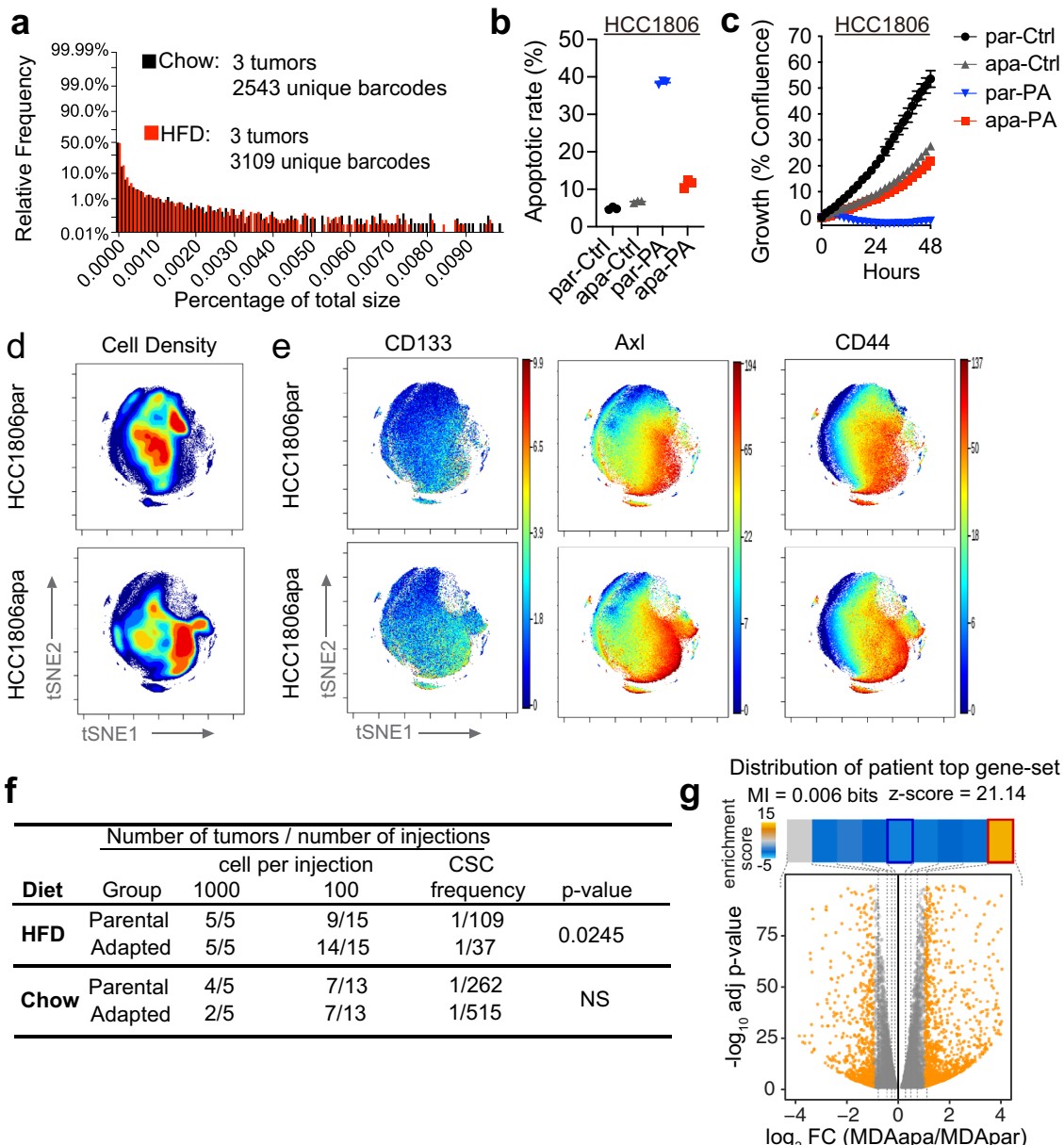

**Fig. 2 Long-term adaptation to PA phenocopy obesity-induced stem-cell features. a** Distributions of relative barcode frequency for three tumors in each group. **b** Apoptotic rate of parental and adapted HCC1806 cells which were treated with 400 µM PA and vehicle (Ctrl) for 48 h. Both early (Annexin V+/PI−) and late (Annexin V+/PI+) apoptotic cells were included for the apoptotic rate % calculation. Data are represented as mean ± SEM of 3 replicates. **c** Time-dependent proliferation assay of parental and adapted HCC1806 following 48 h. Cells were exposed to 400 µM PA and vehicle (Ctrl). Cell growth was determined by high content imaging and represented as % confluence normalized to $t = 0$. For each time point, data are represented as mean ± SEM of 6 replicates. **d** Representative contour plots of mass cytometry data colored by density of cells showing the changes between parental and adapted HCC1806 cells. Color code represents the cell density from low (blue) to high (red). **e** Representative tSNE plots of single parental and adapted HCC1806 cells colored by expression of CD133, Axl, and CD44. **f** Tumor incidence following orthotopic implantation of the indicated number of parental and adapted E0771 cells into chow and HFD-fed mice. The frequency of cancer stem-like cells was calculated by the extreme limiting dilution analysis. The default Chisquare test in ELDA was performed to evaluate the differences between parental and adapted cells (NS, P value >0.05). **g** The distribution of genes induced by obesity (obese and overweight compared to non-obese patients) in PM hormone negative breast cancers patients among the gene expression changes observed in PA-adapted cell lines. We have included the mutual information value (MI) and its associated z-score reported by iPAGE. For visualization, the enrichment/depletion of the query gene-set was determined using the hyper-geometric test and the resulting p-value was used to define an enrichment score that is shown as a heatmap across the expression bins. The obesity-induced genes were significantly enriched in the top-most bin. The red and blue borders in the heatmap denoted statistical significance for enrichment and depletion respectively. Source data are provided as a Source data file.

To obtain insights into the cellular dynamics of cellular adaptation to PA, we performed a single-cell mass cytometry analysis of MDA-MB-231 and HCC1806 parental (MDApar and HCC1806par) and adapted cells, using an antibody panel targeting 27 markers of cellular differentiation states and signaling pathways (Supplementary Table S2). Dimensionality reduction and visualization were performed based on the differential marker expression profiles and cellular densities were

depicted in the tSNE maps. This analysis revealed that PA adaptation governed a clear shift within the cancer cell subpopulations (Fig. 2d and Fig. S2D). Prominently, cellular subpopulations enriched by adaptation to PA were characterized by increased expression of the cancer stem cell markers CD44[34], CD133 and Axl, in both HCC1806 (Fig. 2e) and MDA-MB-231 cells (Fig. S2E). This supports that long-term adaptation to PA phenocopies adaptation to the obese environment by inducing cellular dedifferentiation towards a cancer stem cell-like state. Increased frequency of CD44[high]/CD133[+] cell populations was validated using flow cytometry (Fig. S2F, G).

These findings suggested that PA-adapted cells would have greater tumor initiation capacity in the obese setting. To test this, we implanted E0771 parental (E0771par) and adapted cells at limiting dilutions in the mammary fat pad of obese and non-obese mice and scored tumor formation rates. Interestingly, in vitro adaptation to PA enriched for stemness properties that confer increased tumor formation capacity in vivo in obese, but not lean mice (Fig. 2f). Further, adaptation to PA in MDA-MB-231, HCC1806, and E0771 cells phenocopied enhanced tumor-sphere formation capacity and metabolic reprogramming as demonstrated in the ex vivo E0771 model (Fig. S2H–M). These findings reveal that in vitro adaptation to high concentration of PA phenocopies key obesity-induced tumor initiation phenotypes.

To ascertain how such adaptation resembles what is observed in obese breast cancer patients, we compared the transcriptional alterations observed during in vitro cellular adaptations to PA to the transcriptional changes induced by obesity in PM hormone negative breast cancers patients. To this end, we applied iPAGE, an information-theoretic framework[35], to query how genes induced or repressed in obesity were changed upon adaptation to PA in the in vitro model. For this analysis, genes were first ordered based on their expression changes between MDA-MB-231 parental and adapted cells and were subsequently divided into 10 equally populated bins. We then assessed the distribution of obesity-associated genes across these bins. As shown in Fig. 2g, we observed a significant depletion/enrichment pattern (MI = 0.006 bits, z-score = 21.14). We specifically noted a significant overlap between genes that were induced by the obesogenic state in patients and those upregulated through in vitro adaptation to PA (Fig. 2g). This shared reprogramming of the gene expression landscape suggested that the in vitro long-term adaptation to high abundancies of PA provides clinically relevant information about the molecular drivers of obesity-induced hormone receptor-negative breast cancers. Combined, this suggests that the cellular reprogramming leading to enhanced tumor initiation in obese patients can be governed by long-term adaptation to PA.

**Adaptation to obese environments induces open chromatin linked with C/EBPB occupancy**. Deregulation of metabolic intermediates has recently been tightly linked to epigenetic remodeling and cell fates[36]. To obtain mechanistic insights into obesity-induced cellular adaptations, we therefore next assessed chromatin accessibility by ATAC sequencing (ATACseq) of E0771 cancer cells collected ex vivo after adaptation to lean or obese environments in vivo (Fig. S3A–C). This demonstrated that exposure to an obese environment causes epigenetic remodeling in already malignant cells (329 gain peaks and 1158 loss peaks; Fig. 3a, b, Fig. S3D, E). To ascertain if such remodeling could be related to cancer cell dedifferentiation, we obtained ATACseq data from isolated murine fetal mammary stem cells, basal cells, luminal progenitors and mature luminal cells (GSE116386). We then performed a principal component analysis of these developmental stages and the E0771 tumors isolated from obese and lean mice. This revealed that

adaptation to the obese environment drives significant epigenetic remodeling towards the fetal mammary stem cell state. Compared to the lean state, tumor adaptation to the obese setting resulted in 11.24 (PC1) and 6.01 (PC2) standard deviations closer to the mean principal components of the fetal mammary stem cell state (Fig. 3c and Fig. S3F–H).

To identify the functional consequence of such epigenetic remodeling, we next aggregated changes in chromatin accessibility near putative transcription factor binding motifs to infer differential motif activity and occupancy[37]. To determine potentially common regulators, we included both obesity-induced (ex vivo E0771 derived from HFD compared to chow mice) and PA-specific adaption (MDAapa compared to MDA-par), which similarly caused chromatin remodeling (Fig. S3I–M). Interestingly, this analysis identified the C/EBPB and C/EBPA transcription factors as having the strongest association with the more accessible chromatin in both obesity-induced and PA-adapted cells (Fig. 3d). The differential motif activity was confirmed by using HOMER motif enrichment analysis (Figs. S3N, S3O). As homologs of the C/EBP family transcription factors, C/EBPA and C/EBPB bind to the similar 8-mer canonical TTGCGCAA motif, which is difficult to resolve with motif enrichment algorithms. However, as C/EBPB is robustly expressed and C/EBPA expression is very low in MDAapa cells, we focused on C/EBPB for downstream studies (719-fold difference; Fig. 3e).

C/EBPB has been implicated in determination of cell fate in a variety of tissues, including mammary gland[38]. To examine relevance of epigenetic regulation of C/EBPB accessibility in early mammary gland developmental processes, we examined published single-nuclei ATACseq analysis of murine mammary cells at different developmental stages (GSE125523). Pseudotime and transcription factor motif analysis revealed that the C/EBPB motif was highly enriched in open chromatin in fetal mammary stem cells and other progenitor cells along the luminal lineage but became inactive as the cells became terminally mature luminal cells (Fig. S3P–R). This suggested that epigenetic activation of C/EBPB may be important for the maintenance of mammary stem/progenitor cell fates.

To independently validate enhanced chromatin accessibility for C/EBPB in the obese setting, we performed protein–DNA mapping (Cut&Run sequencing) against activating (H3K4me1) and repressive (H3K27me3) histone marks[39] in MDApar and MDAapa cells, across the same regions (−1 kb - +1 kb relative to C/EBPB motifs) as assessed by ATACseq (Fig. 3f). Cut&Run uses micrococcal nuclease tethered to DNA-bound proteins to generate short DNA cleavage fragments and thus enables base-resolution digital footprints that reflect precise protein–DNA binding sites[40]. Consistent with the ATACseq analysis, the active mark was increased and the repressive mark was repressed in the MDAapa compared to the MDApar cells (Fig. 3g, h). Collectively, the ATACseq and Cut&Run analysis implicate epigenetic activation of C/EBPB transcriptional activity as a major driving factor of tumor-initiating capacity in obese breast cancer.

**C/EBPB promotes tumor stemness specifically in obese environments**. We next asked if C/EBPB is functionally related to obesity-induced cancer stemness. C/EBPB is encoded by an intron-less gene that is expressed in three isoforms; LAP1, LAP2, and LIP by alternative use of transcription start sites[41,42]. Functional depletion of C/EBPB by independent short hairpin RNAs (shRNAs) in E0771apa cells (Fig. 4a) led to a significant reduction in tumorsphere formation capacity (Fig. 4b) without affecting tumor proliferation (Fig. 4c). C/EBPB depletion also resulted in decreased reliance on PA oxidation and reduced use of glucose

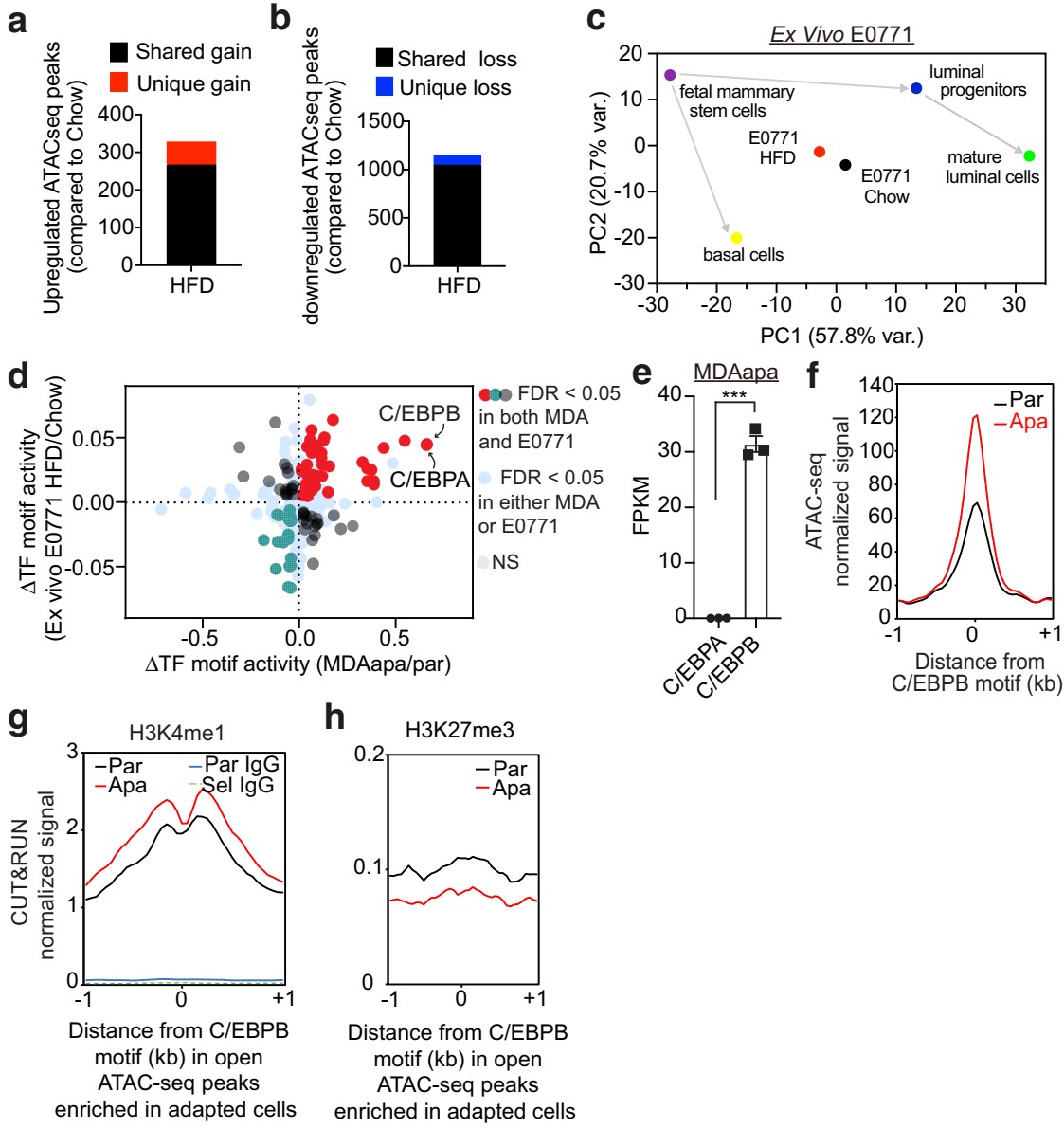

**Fig. 3 Adaptation to obese environments induces open chromatin linked with C/EBPB occupancy. a**, **b** Total number of significantly upregulated (**a**) and downregulated (**b**) ATACseq peaks in E0771 HFD (*n* = 3) relative to chow (*n* = 4) by DiffBind with a false discovery rate (FDR) < 0.05. Unique gain (**a**) or loss (**b**) peaks refer to the peaks identified only in the HFD or chow condition, respectively, whereas shared peaks are peaks called in both conditions. **c** Principal component analysis showing principal components (PC) 1 and 2 of E0771 ex vivo cells and different cell lineages along the mammary gland developmental trajectory (GEO: GSE116386) using the average transcription factor motif activity estimated by chromVar. **d** Overlap of differential transcription factor binding motif activity between MDA-MD-231 (apa/par) and E0771 (HFD/Chow) as determined by diffTF. **e** Within-sample normalized gene expression of transcription factor homologs C/EBPA and C/EBPB in PA-adapted MDA-MB-231 cells using RNA-seq. FPKM = fragments per kilobase of transcript per million mapped reads. Data are represented as mean ± SEM of 3 replicates from each group. Statistical significance determined with unpaired, two-tailed Student's *t*-test (*P* < 0.0001). **f** Metagene representation of the mean ATACseq signal across more accessible C/EBPB motif regions in parental (*n* = 3) or adapted (*n* = 3) MDA-MB-231 cells. The mean signal of three adapted or parental MDA-MB-231 biological replicates was determined by averaging signals of 1 kb around the center of C/EBPB DNA-binding motifs. **g**, **h** Metagene representation of the mean H3K4me1 (par *n* = 2; apa *n* = 2) and IgG signals (**g**) and the mean H3K27me3 signals (par *n* = 3; apa *n* = 3) (**h**) across more accessible C/EBPB motif regions as in (**f**) in MDA-MB-231 cells. The mean signals of biological replicates were determined by averaging signals of 1 kb around the center of C/EBPB DNA-binding motifs. Source data are provided as a Source data file.

for oxidation (Fig. 4d, e) demonstrating that C/EBPB is functionally required for key obesity-induced phenotypes. Importantly, upon transplantation into the mammary fat pad, depletion of C/EBPB significantly delayed the onset of tumor formation in the obese setting, while the knockdown had no effect in the non-obese setting (Fig. 4f). All together, these experiments support a model wherein C/EBPB is associated with transcriptionally active

chromatin and is required for the cancer stem-like phenotype in obesity.

Both LAP1 and LAP2 C/EBPB isoforms contain a dimerization and a transcriptional regulation domain and function as dimers[41]. LIP lacks the DNA-binding domain and has been suggested to function as a competitive inhibitor of LAP1/2[41]. As the protein levels of C/EBPB isoforms and cellular localization did not differ between

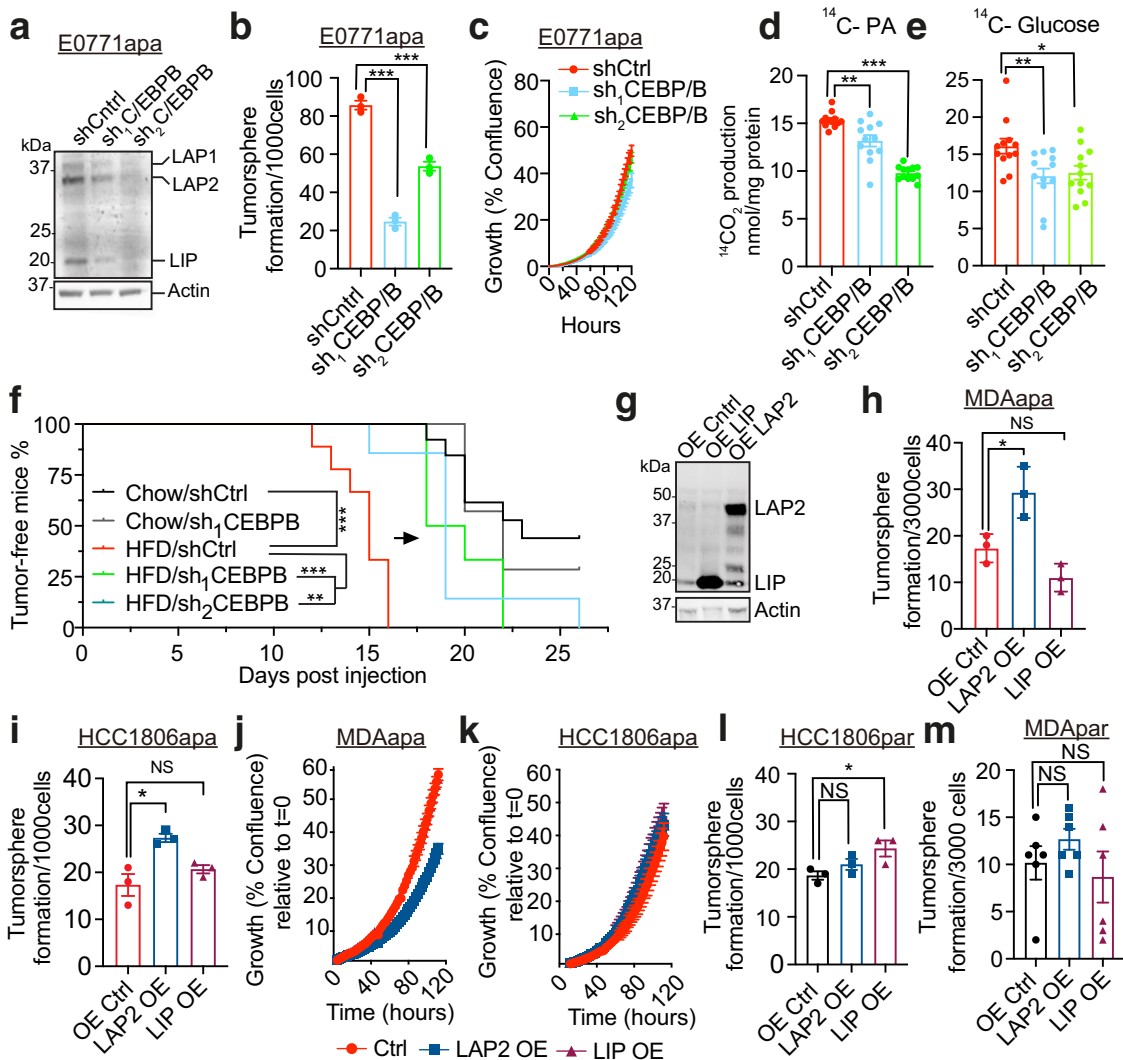

**Fig. 4 C/EBPB promotes tumor stemness specifically in obese environments. a** Western blots of C/EBPB and Actin in cell lysates extracted from knockdown control (shCtrl) and two independent *C/ebpb* knockdown adapted E0771 cells. Three C/EBPB isoforms, LAP1, LAP2, and LIP are marked in the blots. Experiment was independently repeated twice. **b** The changes of tumorsphere formation upon knockdown of *C/ebpb* on adapted E0771 cells. Data are represented as mean ± SEM of 3 replicates. $P_{ctrl/sh1} < 0.0001$, $P_{ctrl/sh2} = 0.0006$. **c** Proliferation assay of control and *C/ebpb* knockdown adapted E0771 cells ($n = 6$ replicates). For each time point, data are represented as mean ± SEM. **d, e** The changes of fatty acid (**d**) and glucose (**e**) oxidation upon *C/ebpb* knockdown in adapted E0771 cells. The oxidation data are normalized to cell protein content ($n = 12$, fatty acid oxidation: $P_{ctrl/sh1} = 0.0028$, $P_{ctrl/sh2} < 0.0001$. Glucose oxidation: $P_{ctrl/sh1} = 0.0097$, $P_{ctrl/sh2} = 0.0165$). **f** Tumor-free survival curves of chow diet and HFD-fed mice orthotopically implanted with E0771 knockdown control and *C/ebpb* knockdown cells. (Chow: shCtrl $n = 13$, $sh_1CEBPB$ $n = 7$; HFD: shCtrl $n = 9$, $sh_1CEBPB$ $n = 6$, $sh_2CEBPB$ $n = 7$; $P_{HFDctrl/sh1} = 0.0004$, $P_{HFDctrl/sh2} = 0.0012$, $P_{chowctrl/HFDctrl} < 0.0001$). **g** Western blots against C/EBPB in cell lysates extracted from control, LAP2 and LIP overexpressed MDA-MB-231 PA-adapted cell line. Actin was used for the normalization. Experiment was independently repeated three times. **h, i** The changes of tumorsphere formation upon the overexpression of C/EBPB LAP2 and LIP isoforms on adapted MDA-MB-231 (**h**) ($n = 3$, $P_{ctrl/LAP2} = 0.0299$, $P_{ctrl/LIP} = 0.0625$) and HCC1806 (**i**, $n = 3$, $P_{ctrl/LAP2} = 0.016$, $P_{ctrl/LIP} = 0.2524$) cells. **j, k** Proliferation assay of control, LAP2 and LIP overexpressed adapted MDA-MB-231 (**j**) and HCC1806 (**k**) cells. For each time point, data are represented as mean ± SEM of 5 replicates and 4 replicates for HCC1806 LAP2 OE. **l, m** The changes of tumorsphere formation upon the overexpression C/EBPB LAP2 and LIP isoforms on parental HCC1806 (**l**) ($n = 3$ $P_{ctrl/LAP2} = 0.1836$, $P_{ctrl/LIP} = 0.0397$) and MDA-MB-231 (**m**) ($n = 6$, $P_{ctrl/LAP2} = 0.2611$, $P_{ctrl/LIP} = 0.6529$) cells. For **b, d, e, h, i, l**, and **m**, statistical significance determined with unpaired, two-tailed Student's *t*-test. For **f**, Log-rank (Mantel–Cox) test was used for statistical testing. (NS, $P$ value >0.05; *$P$ value <0.05; **$P$ value <0.01; ***$P$ value <0.001). Source data are provided as a Source data file.

adapted and parental cells (Fig. S4A–D), we reasoned that obesity- and PA-dependent epigenetic remodeling is required to confer stem-like properties. To test this, we overexpressed either LAP2 (containing a conservative ATG to ATC mutation that eliminate the LIP translation start site) or LIP in both parental and PA-adapted cells (Fig. 4g). Interestingly, ectopic overexpression of the LAP2 isoform C/EBPB in adapted MDA-MB-231 and HCC1806 conferred increased tumorsphere formation capacity, increased frequency of CD44$^{high}$/CD133$^+$ cell populations and metabolic

rewiring without increasing cellular proliferations rates (Fig. 4h–k; Fig. S4E–G). In contrast, equal ectopic overexpression of C/EBPB in parental cells did not affect tumorsphere formation (Fig. 4l, m). Collectively, these findings suggest that epigenetically controlled accessibility of the C/EBPB isoform LAP2 is a key driver of cancer stem-like properties in the obese setting.

**Differential C/EBPB occupancy regulates extracellular matrix organization.** Having shown that C/EBPB is required and

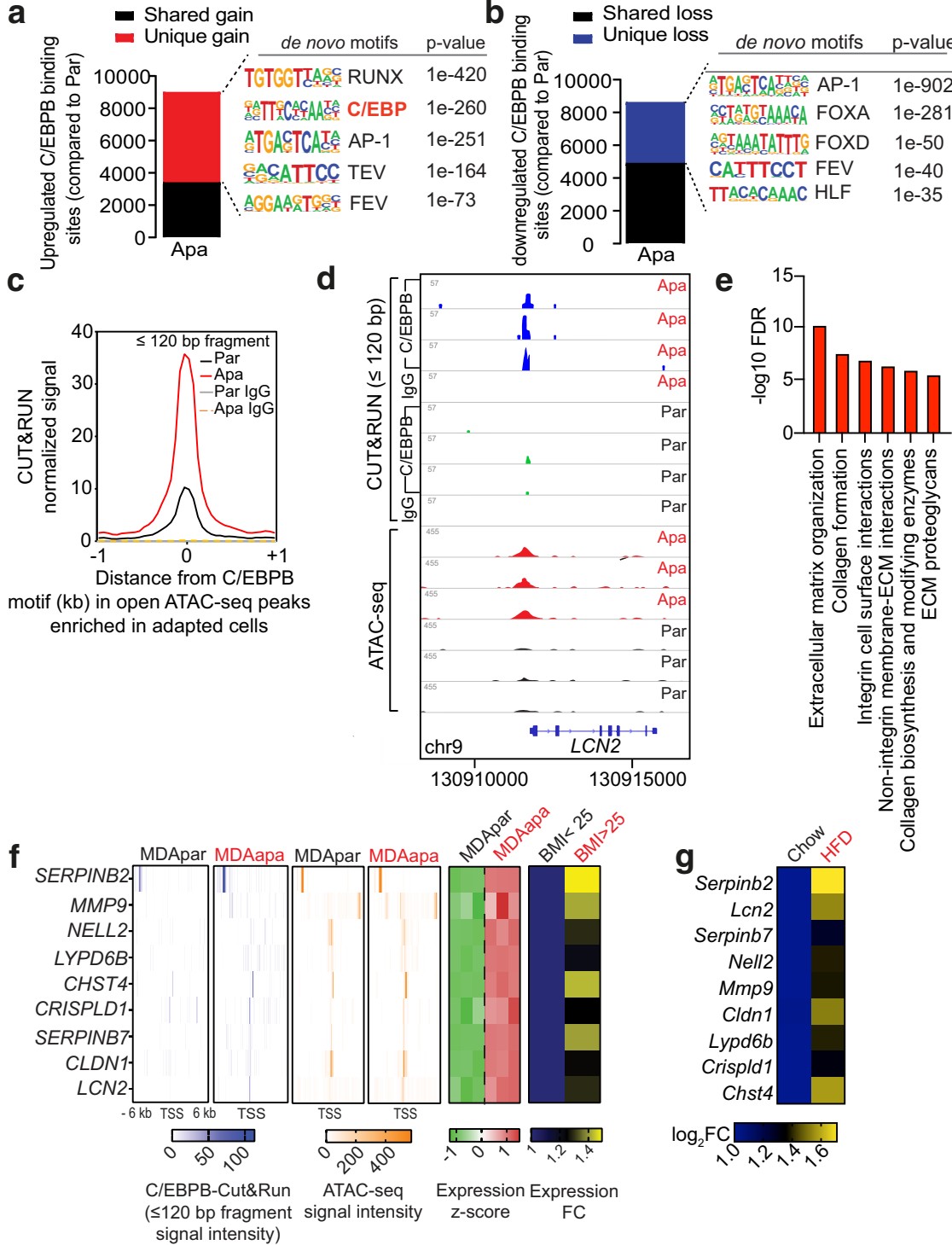

sufficient for tumorsphere formation capacity, we next applied Cut&Run to confirm its genome-wide occupancy and to identify its putative downstream transcriptional targets. Although the total number of upregulated and downregulated C/EBPB peaks were similar in MDAapa relative to MDApar, we observed that a substantial number of these peaks were uniquely present in MDAapa (5618 unique peaks, Fig. 5a, red) and in MDApar (3718 unique peaks, Fig. 5b, blue). De novo motif discovery analysis revealed that the C/EBP family motif was significantly enriched only in the unique peaks in MDAapa, but not those in MDApar (Fig. 5a, b). As an independent means to confirm C/EBPB binding in MDAapa irrespective of peak-calling algorithm, we

enumerated the ends of every Cut&Run fragment (≤120 bp) for each base of the genome and detected significant footprints de novo based on the footprint occupancy score[43]. As expected, motif enrichment analysis identified C/EBPB as the most significantly enriched motif in Cut&Run footprints (Fig. S5A, B). Also, increased C/EBPB binding coincided with the chromatin regions which had increased accessibility in MDAapa cells as compared to MDApar cells (Fig. 5c). These findings confirm increased C/EBPB occupancy in its canonical binding sites in MDAapa, whereas the observed peaks in MDApar may represent nonspecific DNA binding of C/EBPB during its search of accessible motif sites[44].

**Fig. 5 Differential C/EBPB occupancy regulates extracellular matrix organization. a, b** Total number of upregulated (**a**) and downregulated (**b**) C/EBPB binding sites in adapted ($n = 3$) MDA-MB-231 cells relative to the parental ($n = 3$) using DiffBind with an FDR < 0.05 (The $p$-values were determined using default binomial distribution in HOMER). Unique gain or loss sites refer to binding sites identified only in the adapted or parental condition, whereas shared peaks are peaks called in both conditions. Top 5 significant de novo motifs enriched in the unique gain or loss sites were called by HOMER. **c** Metagene representation of the mean C/EBPB Cut&Run signal (fragment length ≤120 bp) across the same chromatin regions as in open ATACseq peak enriched in adapted cells from three biological replicates of adapted or parental MDA-MB-231 cells. Control IgG Cut&Run experiment in adapted and parental cells was included for comparison. **d** Representative genome browser tracks of normalized C/EBPB and IgG Cut&Run and ATACseq profiles around the LCN2 locus in biological replicates of parental and adapted MDA-MB-231 cells. **e** Reactome pathway analysis of genes containing gained chromatin accessibility to C/EBPB. **f** Heatmaps showing average Cut&Run and ATACseq signal intensity centered around the transcription start site (TSS) of the nine putative C/EBPB target genes, and the corresponding mRNA expression of the same genes in three biological replicates of MDApar and MDAapa cells (panels 1–5). Heatmap of expression fold change of the same genes in obese and overweight compared to lean patients was also shown (panel 6). **g** Heat map showing mRNA expression of potential C/EBPB targets in E0771 cells isolated from chow diet and HFD-fed mice. mRNA expression was measured by RT-qPCR with cells isolated from $n = 2$ chow tumors and $n = 3$ HFD tumors. Source data are provided as a Source data file.

To determine putative genes regulated by C/EBPB epigenetic remodeling, we focused on genes whose expression increased and where there were distal or proximal gains in C/EBPB occupancy and chromatin accessibility in PA adaptation. In addition, we considered high-confidence enhancer-gene associations identified cross-platform in GeneHancer[45] (e.g., *LCN2*; Fig. 5d). Pathway analysis of these regions revealed a significant enrichment in processes involved in ECM organization (Fig. 5e), suggesting a link between C/EBPB-dependent ECM remodeling and cancer tumor formation.

To integrate our data derived from the in vivo E0771 obesity model and the in vitro PA adaptation system with the clinical setting, we subsequently focused on the subset of the putative C/EBPB target genes whose expression was significantly elevated in the obese as compared to the lean PM/ER⁻/PR⁻ patients. This analysis identified nine genes, namely, *SERPINB2*, *LCN2*, *SERPINB7*, *NELL2*, *MMP9*, *CLDN1*, *LYPD6B*, *CRISPLD1*, and *CHST4* (Fig. 5f). Interestingly, all of these nine genes had elevated expression in E0771 cells analyzed ex vivo after having been grown in obese and non-obese mice (Fig. 5g). In sum, these data supported a model wherein obesity-induced C/EBPB chromatin binding, activating a transcriptional network involved in ECM regulatory processes.

**CLDN1 and LCN2 are required for C/EBPB-dependent stem cell-like capabilities.** To determine the functional importance of the nine genes in C/EPBP-dependent cancer stemness, we next assessed the levels of the nine genes in cells where C/EBPB was overexpressed. Ectopic overexpression of the LAP2 isoform of C/EBPB in MDAapa and HCC1806apa cells led consistent induction of two genes, *LCN2* and *CLDN1* (Fig. 6a, b and Fig. S6A, B), which paralleled the differential expression patterns observed in cells adapted to obese and non-obese environment (Fig. 5g). Ectopic expression of LIP did not affect expression levels across the potential C/EBPB targets genes (Fig. 6a, b). We therefore functionally tested the role of LCN2 and CLDN1 in tumorsphere formation assays and found that depletion of both genes reduced tumorsphere formation capacity without affecting proliferation rates (Fig. 6c–f, Fig. S6C, D)—thus phenocopying C/EBPB depletion. Further, both LCN2 and CLDN1 were epistatically required for LAP2 induced tumorsphere formation capacity (Fig. 6g, h and Fig. S6E, F). This suggested that CLDN1 and LCN2 were the main downstream mediators of C/EBPB induced stemness. To validate these findings in vivo, we implanted control, CLDN1 and LCN2 depleted cells into the mammary fat pad of obese and lean mice and assessed tumor formation. While high-fat feeding resulted in accelerated tumor formation rates, this was prevented by depletion of either CLDN1 or LCN2 (Fig. 6i, j). Supportive of an obesity-specific effect, depletion of either CLDN1 or LCN2 did not affect tumor take rate in the lean

mice (Fig. 6I, J). Combined, these results suggest that CLDN1 and LCN2 are the downstream mediators of C/EBPB induced tumor formation capacity in the obese setting.

## Discussion

Obesity is a complex pathological condition that conceivably affects the formation and development of cancers through multiple avenues. Here we have demonstrated that cancer cell exposed to the obese environments specifically adapts to high levels of PA to drive enhanced tumor formation capacity in PM/ER⁻/PR⁻ breast cancer. Our findings further suggest that this is mediated through a general cellular adaptation process rather than expansion of a pre-existing cellular subpopulation. We find that obesity-induced adaptation to PA governs dedifferentiation of cancer cells towards a tumor stem cell-like phenotype leading to augmented tumor formation capacity. Clinically, this manifests in a higher cancer cell frequency of CD133⁺ and Axl^high cancer stem cells and shorter disease-specific survival in obese and overweight PM/ER⁻/PR⁻ breast cancer patients compared to normal weight patients. This is corroborated epidemiologically by the association of obesity with higher cancer risk[46] and poor prognosis[4] of PM/ER⁻/PR⁻ breast cancer patients[47].

Our studies did not identify any specific genetic mutations correlating to obesity in the tumors from PM/ER⁻/PR⁻ breast patients. Such lack of a genetic link between obesity and cancer formation were supported in a mutated *Kras*-dependent pancreatic ductal adenocarcinoma model. Here, obesity, as induced through the *ob/ob* mutation, led to enhanced tumorigenicity independently of the acquisition of new driver mutations[48]. In contrast to a genetic link, we identified a critical link between adaptation to the obese environment and genome-wide changes in chromatin accessibility. This is supported by recent observations that high-fat feeding leads to alterations in chromatin interactions to drive adaptive networks in the liver[49]. These interactions might reflect diet-induced alterations in metabolic intermediates that are intimately connected to epigenetic control of gene transcription[50,51]. Interestingly, lipid-derived acetyl-CoA has been suggested to be the source of up to 90% of acetylation modifications of certain histone lysine's[52]. Our work and the recent work by Ringel et al.[7], both describe obesity-dependent changes in lipid handling in cancer cells. This suggests that alteration to lipid-derived acetyl-CoA could potently affect the chromatin landscape of cancer cells and thus link obesity to tumor formation and progression.

Importantly, we demonstrate that the obesity-dependent epigenetic remodeling is specific, rendering chromatin regions containing the binding motif for C/EBPB more accessible and thereby activating a C/EBPB-dependent transcriptional network. Through complementary sets of in vitro and in vivo experiments, we show that C/EBPB is required for obesity-induced tumor

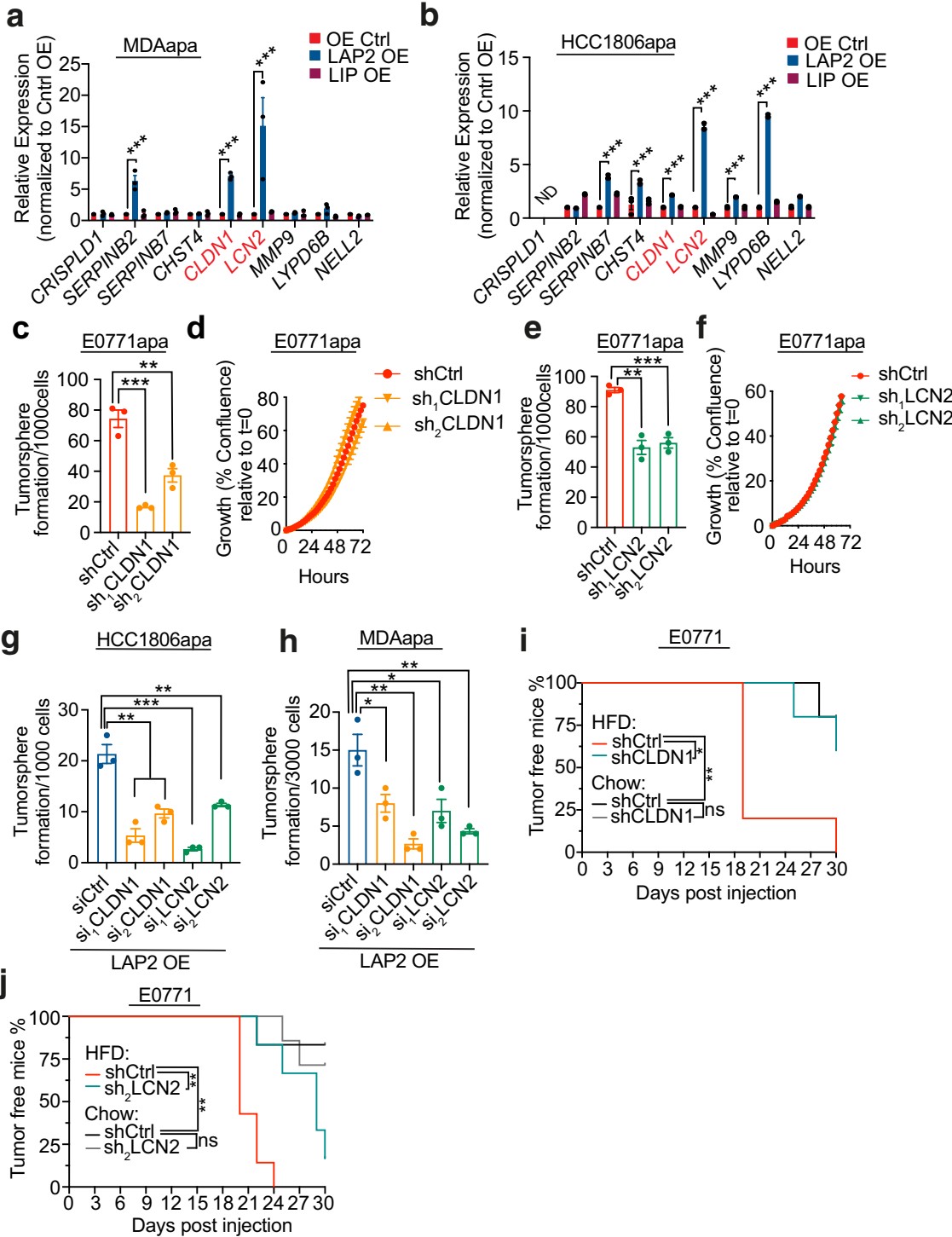

formation. Conversely, ectopic overexpression of C/EBPB enhanced the frequency of cancer stem cells. Previous reports observed that C/EBPB is required for stem cell maintenance in the developing breast using whole-body knockout mice[38] and that expression of the LAP2 isoform of C/EBPB can transform a non-cancerous cell line MCF10A[53], lending further support to the functional role for C/EBPB-dependent tumor formation.

Our unbiased Cut&Run analysis of direct C/EBPB target genes suggested that C/EBPB regulates tumor initiation features through regulation of the surrounding ECM. Cancer cell-autonomous regulation of the ECM is intrinsically linked to cancer stemness through manipulation of mechanical properties and signaling molecules[54,55]. Consistent with our findings, obesity-induced alterations in the ECM mechanics have been reported to support tumorigenesis[56]. Interestingly, a total of nine C/EBPB target genes were also induced in obese PM/ER−/PR− breast cancer patients. Of these nine genes, depletion CLDN1 and LCN2 phenocopied C/EBPB knockdown and were epistatically required for C/EBPB induced tumorsphere formation capacity in vitro and tumor formation rate in vivo suggesting that these engender the downstream effects of C/EBPB (Fig. 7).

LCN2 is induced in adipose tissue of obese individuals[57] and was previously described to induce inflammation and fibrosis and in an obesity-driven pancreatic ductal adenocarcinoma model[58].

**Fig. 6 CLDN1 and LCN2 are required for C/EBPB-dependent stem cell-like capabilities. a, b** The changes in the expression of C/EBPB potential target genes upon the overexpression of LIP and LAP2 on adapted MDA-MB-231 (For SERPINB2 adjust $P = 0.0002$, and for CLDN1 and LCN2 adjust $P < 0.0001$) (**a**) and HCC1806 (adjust $P < 0.0001$, ND = not detectable) (**b**) cells. The expression of target genes is shown as relative fold change over Control OE. Data shown as mean ± SEM of 3 independently repeated experiments, two-way ANOVA multiple comparisons were performed to assess statistical significance. **c** The changes of tumorsphere formation of *Cldn1* depletion in E0771apa cells ($n = 3$; $P_{ctrl/sh1} = 0.0006$, $P_{ctrl/sh2} = 0.0068$). **d** Proliferation of control and *Cldn1* knockdown adapted E0771 cells. For each time point, data are represented as mean ± SEM of 8 replicates. **e** The changes of tumorsphere formation of *Lcn2* depletion in E0771apa cells ($P_{ctrl/sh1} = 0.0015$, $P_{ctrl/sh2} = 0.0008$). **f** Proliferation of control and *Lcn2* knockdown E0771apa cells. For each time point, data are represented as mean ± SEM of 8 replicates. **g, h** The changes of tumorsphere formation upon knockdown of CLDN1 and LCN2 with two independent siRNAs on the LAP2 overexpressed adapted HCC1806 ($n = 3$; $P_{ctrl/si1CLDN1} = 0.0022$, $P_{ctrl/si2CLDN1} = 0.0047$, $P_{ctrl/si1LCN2} = 0.0006$, $P_{ctrl/si2CLDN1} = 0.0061$) (**g**) and MDA-MB-231 ($n = 3$; $P_{ctrl/si1CLDN1} = 0.0424$, $P_{ctrl/si2CLDN1} = 0.0049$, $P_{ctrl/si1LCN2} = 0.0363$, $P_{ctrl/si2CLDN1} = 0.0072$) (**h**) cells. **i** Tumor-free survival curves of chow and HFD-fed mice orthotopically implanted with 100 E0771 control and *Cldn1* knockdown cells ($n = 5$ in each condition, $P_{HFDctrl/sh1} = 0.0173$, $P_{HFDctrl/Chowctrl} = 0.0074$). Tumor volume was measured every 2–3 days and tumor formation were recorded when reached a volume 50 mm$^3$. **j** Tumor-free survival curves of chow diet and HFD-fed mice orthotopically implanted with 100 E0771 control and *Lcn2* knockdown cells. The analysis was performed by using the mice from two independent experiments (HFD/shCtrl $n = 7$, HFD/sh$_2$LCN2 $n = 6$; Chow/shCtrl $n = 6$, Chow/sh$_2$LCN2 $n = 7$, $P_{HFDctrl/sh1} = 0.0018$, $P_{HFDctrl/Chowctrl} = 0.0018$). For **c, e, g, h**, data shown as mean ± SEM of 3 replicates, and statistical significance determined with unpaired, two-tailed Student's *t*-test. For **i, j**, $P$ values were determined with Log-rank (Mantel–Cox) test (NS, $P$ value >0.05; *$P$ value <0.05; **$P$ value <0.01; ***$P$ value <0.001). Source data are provided as a Source data file.

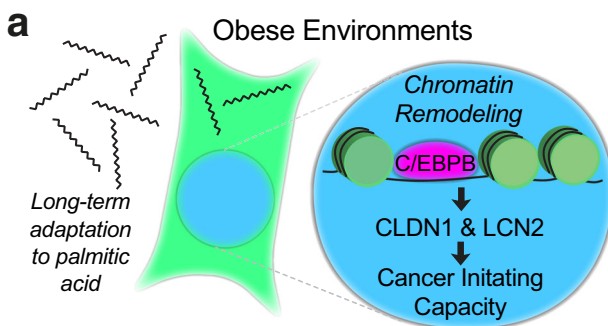

**Fig. 7 A schematic model of obese environment on breast cancer-initiating capacity.** Long-term adaptation of breast cancer cells to palmitic acid promotes tumor-initiating capacity through increased accessibility of C/EBPB binding motifs, which induces the expression of C/EBPB targets *CLDN1* and *LCN2*.

In breast cancer, LCN2 has been linked to cellular differentiation through modulation of the epithelial to mesenchymal transition[59]. As a tight junction protein, CLDN1 is expressed in several types of human cancers[60–62]. In breast cancers, CLDN1's expression was significantly associated with the basal-like subtype of breast cancers[61]. Accumulating evidence has demonstrated that CLDN1 induces EMT to lead metastatic behavior in colon[60] and liver cancer[62]. As such, both CLDN1 and LCN2 have been suggested to be involved in cancer dedifferentiation, future work is needed to establish the mechanistic basis of their actions—especially in the context of obese environments.

Aberrant lipid metabolism is a hallmark of deregulated cancer metabolism[63]. It has been widely reported that cancer cells augment their de novo lipid biosynthesis for energy production, synthesis of new membranes, to regulate membrane structures that coordinate signal transduction, and for the biosynthesis of lipid signaling molecules such as phosphatidylinositol-3,4,5-trisphosphate[64]. In addition, cancer cells can stimulate the release of fatty acids from surrounding adipocytes to provide energy for tumor growth[65]. In support of a link between fatty acids and stemness, is the observation that slow-cycling metastasis-initiating cells are dependent on the lipid uptake protein CD36[66]. While we did not observe any direct involvement of CD36 in our studies of obesity-induced breast cancer, both studies describe a critical role for fatty acid metabolism in cancer stemness. Specifically, our study expands on the importance of fatty acids, by demonstrating that obesity-induced PA concentrations drives cellular adaptation

of the cancer cell suggesting that PA might exert a critical regulatory role outside of its role in energetics during tumor formation.

Combined, our analysis of cellular adaptations to obese environments has revealed changes of cellular phenotypes, driven by the combined modulation of C/EBPB transcriptional activity. In the context of personalized medicine, this suggests that obese cancer patients might benefit from specific targeted therapies rather than generic treatment regiments.

## Methods

**Breast cancer patient cohort.** This study enrolled a total of 223 patients with primary stage III breast cancers. Out of these 115 patients were PM patients (defined by age >50 years). Recruitment period was between November 24, 1997 and December 16, 2003. The median age was 51 years (range 25–70). Patients' BMI, age, hormone status at the time of diagnosis, as well as patient survival times (overall survival and disease-specific survival) were documented. The study was approved (including informed consent) by the regional committees for medical and health research of Western Norway (REK-Vest; approval number 273/96-82.96). More details about the study cohort can be found in the following report[15].

**Animal models.** All animal experiments were approved by the Norwegian Animal Research Authority and conducted according to the European Convention for the Protection of Vertebrates Used for Scientific Purposes, Norway. The Animal Care and Use Programs at University of Bergen are accredited by AAALAC international. The laboratory animal facility at University of Bergen was used for the housing and care of all mice. C57BL/6J mice were obtained from Jackson Laboratories and bred on site. Female mice were kept in IVC-II cages (SealsafeÒ IVC Blue Line 1284L, Tecniplast, Buguggiate, Italy); 5–6 mice were housed together and maintained under standard housing conditions at 21 °C ± 0.5 °C, 55% ± 5% humidity, and 12 h artificial light-dark cycle (150 lux). Mice were provided with standard rodent chow (Special Diet Services, RM1 801151, Scanbur BK, Oslo, Norway) and water ab libitum.

To mimic both obese and non-obese environments, 6-week-old female littermates were randomly assigned to chow and HFD groups and fed either standard chow diet (7.5% kcal from fat, 17.5% from proteins and 75% from carbohydrates, Special Diet Services RM1, 801151) or high fat containing diets (60% kcal from fat, 20% from protein and 20% from carbohydrates, Research Diets, D12492) for 10 weeks prior to tumor cell implantations. Body weight was monitored every week. The respective diets were maintained throughout the experiment.

**Cell lines and culture.** MDA-MB-231 (TNBC, human), HCC1806 (TNBC, human), and HEK293T cell lines were purchased from the American Type Culture Collection (ATCC). E0771 (TNBC, mouse) cell line was purchased from the CH3 BioSystems. TeLi (basal breast cancer, mouse) cells were originally derived from a tumor formed in MMTV-Wnt1 transgenic mouse and then propagated in vivo for four generations through mammary fat pad injections before being passaged in vitro. Tumors were dissociated using Mouse tumor dissociation kit (Miltenyi Biotec, 130-096-730) according to manufacturer's instructions. Dissociated tumor cells were cultured in vitro for 2 months to obtain pure tumor cells. The in vivo passaged MMTV-Wnt cells were kindly provided by Stein-Ove Døskeland, University of Bergen. MDA-MB-231, E0771, and TeLi cells were cultured at 37 °C, 5%

$CO_2$ in high-glucose DMEM (Sigma, D5671) supplemented with 10% FBS (Sigma, F-7524), 100 U/mL penicillin and 100 μg/mL streptomycin (Sigma, P-0781) and 4 mM L-glutamine (Sigma, G-7513). HCC1806 cells were cultured in RPMI1640 (Sigma, R8758) supplemented with 10% FBS and 100 U/mL penicillin/and 100 μg/mL streptomycin.

For cell line authentication, MDA-MB-231 cells were harvested for genomic DNA extraction using Genomic DNA isolation kit (Norgen Biotek, 24700). Isolated genomic DNA was analyzed by Eurofins Genomics laboratory and the cell line authenticated based on genetic fingerprinting and short tandem repeat (STR) profiling.

**Patient tissue microarray and transcriptomic analysis**. Tissue specimens were from the human breast cancer patient cohort described above[15]. At the time of diagnosis, each patient from the study cohort had an incisional tumor biopsy. All tissue samples were fixed in formaldehyde for paraffin embedding in the operating theater immediately after removal. Paraffin-embedded tissues were subject to tissue microarray (TMA) construction. From each tumor, 4 cores of 1.2 mm diameter from tumor-rich areas were punched out using Manual Tissue Arrayer Punchers (MP10; Beecher Instruments). The patient cores were embedded into ten $8 \times 10$ array blocks plus 1 to 2 liver control cores for orientation. Microtome sectioned slides were stored at 4 °C until ready for use.

Immunohistochemical staining for CD133 was done as described previously[17]. In short, slides were dried at 58 °C over two days and deparaffinization was performed using xylene, rehydrated with ethanol and $dH_2O$. Target retrieval was done in Tris/EDTA buffer, pH 9 (Dako, S2367) in a microwave for 25 min. Slides with buffers were cooled down at room temperature for 15 min, followed by rinsing with cold $dH_2O$. Samples were then blocked in the Peroxidase Blocking solution (Dako REAL, S2023) for 8 min, rinsed with water, and then blocked in a serum-free protein block buffer for 8 min (Dako, X0909). Primary CD133 antibody (Miltenyi Biotec, 130-090-422) was diluted 1:25 in Antibody Diluent with Background Reducing Components (Dako, S3022). 200 μL of antibody solution was put on each slide to cover all TMA specimens and incubated overnight at 4 °C. The following day, slides were washed twice with Dako Wash Buffer (S3006). Primary antibody detection was performed using MACH3 mouse probe (Biocare Medical) followed by MACH3 HRP polymer (Biocare Medical, BC-M3M530H) and the signal was developed with diamino-benzidine DAB+ (Dako, K3468). Finally, the slides were counterstained with hematoxylin (Dako, S3301), dehydrated in alcohol solutions and xylene, and mounted in Pertex Mount Agent (Histolab, 00801).

Immunohistochemical staining for Axl was done as described previously[67]. In short, slides were dried at 58 °C over two days and deparaffinization was performed using xylene, rehydrated with ethanol and $dH_2O$. Target retrieval was done in Tris/EDTA buffer, pH 6 (Dako, S2367) in a microwave for 25 min. Slides with buffers were cooled down at room temperature for 15 min, followed by rinsing with cold $dH_2O$. Samples were then blocked in the Peroxidase Blocking solution (Dako REAL, S2023) for 15 min, rinsed with water, and then blocked in a serum-free protein block buffer for 15 min (Dako, X0909). Primary Axl antibody (RnD Systems, AF154) was diluted 1:400 in Antibody Diluent with Background Reducing Components (Dako, S3022). 200 μL of antibody solution was put on each slide to cover all TMA specimens and incubated overnight at 4 °C. The following day, secondary Rabbit anti-Goat IgG control antibodies were added on the slides in 1:400 dilution in DAKO Antibody Diluent for 30 min at room temperature. Further, slides were washed twice with Dako Wash Buffer (S3006). Secondary antibody detection was performed using DAKO EnVision (DAKO, K4003) followed by the signal development by diamino-benzidine DAB+ (Dako, K3468). Finally, the slides were counterstained with hematoxylin (Dako, S3301), dehydrated in alcohol solutions and xylene, and mounted in Pertex Mount Agent (Histolab, 00801).

mRNA expression levels were extracted from previously reported microarray analyses[68]. In brief, these analyses were performed on a Human HT-12-v4 BeadChip (Illumina) after labeling (Ambion; Aros Applied Biotechnology). Illumina BeadArray Reader (Illumina) and the Bead Scan Software (Illumina) were used to scan BeadChips. Expression signals from the beads were normalized and further processed as previously described[69]. The data set was re-annotated using illuminaHumanv4.db from AnnotationDbi package, built under Bioconductor 3.3 in R[70], to select only probes with "perfect" annotation[71]. The probes represented 21043 identified and unique genes.

Targeted sequencing of 360 cancer genes, was performed and described previously[28]. In brief, native, genomic DNA from tumor was fragmented and subjected to Illumina DNA sequencing library preparation. Libraries were then hybridized to custom RNA baits according to the Agilent SureSelect protocol. Paired-end, 75 bp sequence reads were generated. Sequencing coverage for the targeted regions (average per bp) within each sample was >120x for all samples (mean 439x). Supplemental Table 1 lists the included 360 genes.

**Proliferation assay**. Cell proliferation assay was determined by high-content imaging using the IncuCyte Zoom (Essen Bioscience) according to the manufacturer's instructions. In all experiments, cells were seeded into a 96-well culture plate and for each well four fields were imaged under ×10 magnification every 2 h. The IncuCyte Zoom (v2018A) software was used to calculate confluency values.

Cell growth was determined by high content imaging and represented as % confluence normalized to $t = 0$.

**Glucose and insulin measurements**. For glucose and insulin measurements, mice were fasted overnight (9 h) with free access to water. Blood glucose concentrations were determined using Accu-Check Aviva glucometer (Roche). For insulin measurements, blood was collected from the tail using EDTA-coated capillary tubes (Fisher Scientific, 11383994), stored on ice before centrifuged at $2000 \times g$, 4 °C for 10 min. Plasma insulin concentrations were determined in duplicates using the Ultra Sensitive Mouse Insulin ELISA Kit (Crystal Chem, 90080) following the manufactures instructions for wide range of measures.

**Glucose tolerance test**. For glucose tolerance test, mice fed a HFD or chow-diet for 10 weeks were fasted overnight (15 h) with free access to water. Glucose (2.5 g/kg) was administered by gavage, and blood glucose concentrations were determined by using Accu-Check Aviva glucometer (Roche).

**Mammary fat pad implantations**. E0771 or TeLi cells were prepared in PBS and mixed 1:1 by volume with Matrigel (Corning, 356231) and orthotopically implanted into the fourth inguinal mammary fat pad of chow and HFD-fed mice in a total volume of 50 μL. Tumor diameters (width and length) were measured 2–3 times per week with caliper. Tumor volumes were calculated using formula tumor volume $(mm^3)$ = Width × $Length^2$ × π/6. Tumors were considered established when the volumes were larger than 50 $mm^3$.

**Cellular adaptation to PA**. Cells were seeded on 10 cm culture dishes so that the confluency at the starting day of adaptation was 80–90%. To start adaptation, all media was removed and replaced by growth media supplemented with palmitic acid (PA) (Sigma, P5585) and 1% fatty acid-free BSA as a carrier (Sigma, A7030). Adaptation was done using gradual increase in PA concentration MDA-MB-231 (50 μM, 200 μM and 400 μM), HCC1806 (200 μM and 400 μM), E0771 (200 μM, 400 μM, 500 μM) ensuring around 50% of cell death at each step. Parental cells were cultured in parallel using growth media supplemented only with 1% fatty acid-free BSA (Sigma, A7030).

For PA supplemented media, PA was first dissolved in absolute ethanol to obtain a 50 mM stock. To prepare the working concentrations, certain volumes of PA stock were added into 1%BSA growth media and incubated at 37 °C for 1 h. PA stock was stored at 4 °C and used for no longer than 2 weeks.

**Generation of knockdown and overexpressing cell lines**. Short hairpin RNAs (shRNA) for target genes and scramble (shCtrl) were purchased from Sigma. pBabe-puro plasmids containing human C/EBPB LAP2 and LIP isoforms were from Addgene (Cat.# 15712 and 15713).

For production of virus, HEK293T cells were seeded onto 10 cm plates to reach 80% confluency on the following day. For retroviral overexpression, 12 μg of Gag/Pol plasmid, 6 μg of VSVG plasmid, and 12 μg of pBabe-puro plasmid containing C/EBPB isoforms were respectively co-transfected into the HEK293T cells using 60 μL Lipofectamine 2000 according to manufacturer's protocol. For lentiviral-mediated depletion of target genes, cells were transfected with 12 μg Gag/Pol plasmid, 6 μg envelope plasmid, and 12 μg shRNA containing plasmid (pLKO).

Six hours following transfection, the media was replaced with fresh media. The virus was harvested 48 h post transfection by spinning the collected culture media for 5 min at 1200 rpm and then filtered through a 0.22 μm filter to completely remove cell debris. The virus was then stored at −20 °C for several days or at −80 °C for several months.

To infect target cells, 5 mL of the appropriate virus was used to infect a subconfluent 10 cm cell culture dish in the presence of 10 μg/mL of polybrene overnight. Forty-eight hours after infection, puromycin was added to select for successfully infected cells: 4 μg/mL for TeLi, 2 μg/mL for MDA-MB-231 and E0771, and 1.33 μg/mL for HCC1806 cells. Uninfected control cells were processed the same way to determine the endpoint of selection. Typically, selection took 2–3 days for all cell lines. After the end of selection, cells were released from puromycin for at least 1 day before starting experiments.

**Tumorsphere formation assay**. Assay was performed as previously described[13,22,72]. Cells were harvested using Accutase and re-suspended in PBS. After counting, indicated number of cells (1000 cells/well for E0771 and HCC1806 cell lines, 3000 cells/well for MDA-MB-231 cell lines) were seeded into ultra-low attachment 6-well plates (Corning) in the stem cell media (DMEM/F12 with 20 ng/mL EGF, 20 ng/mL bFGF, 1x B27 supplement). Tumorspheres were quantified after 5 days (for E0771 cell line) or 7–10 days incubation (for HCC1806 and MDA-MB-231 cell lines). Tumorspheres were counted when the size is larger than 60 μm for HCC1806 and MDA-MB-231 cells or 100 μm for E0771 cells. For tumorsphere propagation assay, tumorspheres were harvested and trypsinized by using Accutase to obtain single cells. Cells were counted and seeded (500 cells/well for E0771 cells and 1000 cells/well for HCC1806 cells) into ultra-low attachment 6-well plates in the stem cell media. Tumorspheres were imaged and quantified after 5–7 days.

**Apoptosis**. Analysis of apoptosis was performed using Alexa Fluor™ 488 conjugate Annexin V (Thermo Fisher, A13201) and propidium iodide (PI) according to the manufacturer's instructions. Shortly, cells and their culture media were harvested and washed once in cold PBS. Cells were then resuspended in Annexin binding buffer (10 mM HEPES, 140 mM NaCl, and 2.5 mM CaCl₂, pH 7.4) in a concentration of $1 \times 10^6$ cells/mL. To each 100 µL of cell suspension 5 µL of the Annexin V and 2 µL PI (at final concentration 2 µg/mL) was added. Cells were incubated in the dark at room temperature for 15 min. After the incubation period, 400 µL of Annexin binding buffer was added and cells were analyzed by flow cytometry (BD LSR Fortessa).

**Flow cytometry analysis**. For immunostaining for flow cytometry, cells were collected using Accutase (Sigma, A6964) and washed once in PBS. $1 \times 10^6$ cells per sample were stained with 0.6 µL of APC conjugated CD133 antibody (Invitrogen, 17-1331-81) and 1 µL of FITC conjugated CD44 antibody (BioLegend, 338803) in 100 µL of 1% BSA supplemented PBS solution and incubated in dark for 20 min at room temperature. After incubation, cells were washed once with 5 mL of PBS/1% BSA and analyzed on flow cytometry (BD LSR Fortessa). To gate the CD44$^{high}$/CD133$^+$ cell population, the median fluorescent intensity (MFI) of CD44-FITC was measured on control replicates (termed parental cells in Fig. S2F, G and overexpression control cells in Fig. S4E, F). The average value of CD44-FITC MFI was used to gate CD44$^{high}$ cells, and CD133$^+$ cells were gated according to the negative staining samples. Gating and cell quantification were performed using BD FACSDIVA, FlowJo.

**Immunofluorescent analysis**. Cells were seeded in 24-well plates on Poly-L-lysin treated cover slips at 75,000 cells per well 1 day before the staining. On the day of the analysis, culture media was removed and 4% paraformaldehyde (PFA) in Distilled-PBS (DPBS) was added to fix cells for 20 min. Then PFA was removed and cells were permeabilized in 0.4% Tween/DPBS for 10 min at RT. This was followed by 3 washes in DPBS. Blocking was performed in 3%BSA/0.2% Tween/DPBS for 90 min. Slides were shortly washed in staining media containing DPBS/0.2% Tween/1.5% BSA. Then slides were covered by 500 µL of staining media with C/EBPB antibodies (1:100 dilution) and incubated overnight at 4 °C. Next day, slides were washed in DPBS 3 × 5 min and incubated with secondary antibodies (1:500 dilution) for 2 h. This was followed by 5 min wash in DPBS, then 5 min incubation with DAPI (1:500 in DPBS), and then another wash in DPBS. Further, slides were rinsed in distilled water and mounted with ProLong™ Diamond Anti-fade Mountant. Slides were dried overnight and imaged using Leica SP5 with ×63 magnification. Image quantification was performed using Fiji software. The nucleus and whole cell were demarcated based on DAPI and bright field, respectively. % nuclear C/EBPB were calculated by diving the nuclear signal by whole-cell signal multiplied by 100.

**Fatty acid and glucose oxidation assay**. Fatty acid and glucose oxidation were assessed by providing $^{14}$C-labled palmitic acid or glucose to the cells, with subsequent capture of the released $^{14}CO_2$; a technique previously described[73]. In brief, cells were plated in 96-well tissue culture plates (MDA-MB-231, 45,000 cells/well; HCC1806, 45,000 cells/well; dissociated E0771, 25,000 cells/well) in corresponding growth medium and incubated overnight to allow proper attachment. Radiolabeled [1-$^{14}$C] palmitic acid (1 µCi/mL) and D-[$^{14}$C(U)] glucose (1 µCi/mL) were given in PBS supplemented with 10 mM HEPES and 1 mM L-carnitine. Respective amounts of non-radiolabeled substrate were added to obtain final concentrations of D-glucose (5 mM) and BSA-conjugated palmitic acid (100 µM). Etomoxir (40 µM) was added to certain wells during palmitic acid oxidation, to monitor the non-mitochondrial $CO_2$ production. An UniFilter®-96w GF/B microplate was activated for capture of $CO_2$ by the addition of 1 M NaOH (25 µL/well) and sealed to the top of the 96-well tissue culture plates and incubated for the indicated period of time at 37 °C. Subsequently, 30 µL scintillation liquid (MicroScint PS PerkinElmer) was added to the filters and the filter plate was sealed with a TopSealA (PerkinElmer). Radioactivity was measured using MicroBeta2 Microplate Counter (PerkinElmer). Protein measurement was performed for data normalization. The cells were washed twice with PBS, lysed by 0.1 M NaOH, and protein was measured using Pierce® BCA Protein Assay Kit (Thermo Fisher Scientific, 23225).

**RNA extraction, RT-PCR, and qPCR**. Total RNA was extracted with a Total RNA purification Kit (NORGEN Biotek, 37500) according to the manufacturer's protocol. cDNA was synthesized from 1 µg total RNA template with oligo-dT primers using a SuperScript® III First-Strand Synthesis kit (ThermoFisher Scientific, 18080-051) according to the manufacturer's protocol. qPCR was carried out in quadruplicates with a LightCycler® 480 SYBR Green I Master Mix (Roche, 04887352001) using a LightCycler® 480 Instrument II (Roche, 05015243001). The results were calculated by ΔΔCt method using human HPRT (h*HPRT*) for human genes and mouse actin (m*Actin*) for mouse genes. Primer sequences are listed in Supplementary Table 4.

**Transfection of siRNA duplexes**. One day before transfection cells were plated on T25 flasks at the density 250,000 cells/flask. After overnight incubation, cells were transfected using Lipofectamine® RNAiMAX according to the manufacturer's

protocol with some modifications: we used 3 µL of Lipofectamine per flask and final concentration of siRNAs was 20 nM. After 48 h incubation cells were harvested and seeded for tumorsphere formation assay.

**Western blotting**. Cells were lysed in RIPA lysis buffer (Thermo Scientific, 89901) complemented with protease inhibitor cocktail (cOmplete ULTRA Tablets, MINI, EDTA-free, EASYpack, 05892 791001) and phosphatase inhibitor cocktail (Phos-Stop, 04906837001). After quantification with the Pierce® BCA Protein Assay Kit, equal amounts of protein (typically 20–50 µg of protein per lane) were separated by electrophoresis on a NuPAGE 10% Bis-Tris Gel (Invitrogen, NP0315BOX) in NuPAGE™ MOPS SDS Running Buffer (×20, Invitrogen, NP000102) and then transferred to an activated Immobilon-P PVDF Membrane (Merck Millipore Ltd, IPVH00010 PORE SIZE: 0.45 µm). After quantification, the membranes were blocked using 5% nonfat dry milk in PBS/0.1% Tween20 for 1 h at RT, incubated with indicated primary antibodies for overnight at 4 °C. This step was followed by an incubation with secondary IRDye-conjugated antibodies (Leicor, P/N 925-68070, P/N 926-32213). Detection and quantification were performed on Amersham Typhoon Gel and Blot Imaging Systems. A list of antibodies is given in the key resources table.

**RNA sequencing**. MDA-MB-231 parental and selected cells were plated at $1 \times 10^5$ cells/mL into 6-well plates in the corresponding medium. After 3 days, cells were harvested, and RNA extraction was performed with a Total RNA purification Kit according to the manufacturer's protocol. Potential DNA contaminations were removed by applying the RNA Clean & concentrator with DNaseI kit (Zymo, R1013). RNA sequencing libraries were prepared at the Genomic Core Facility at University of Bergen using Illumina TruSeq Stranded mRNA sample preparation kit according to the manufacturer's instructions and sequenced on the same lane on a HiSeq 4000 sequencer with paired-end 75 bp reads.

**ATACseq library construction**. ATACseq libraries were constructed as previously described[74]. In brief, $5 \times 10^4$ cells were washed once with ice-cold PBS and pelleted by centrifugation. Cells were lysed in 50 µL RSB buffer (10 mM Tris-HCl pH 7.4, 10 mM NaCl and 3 mM MgCl₂) containing 0.1% NP-40, 0.1% Tween-20 and 0.01% digitonin, and incubated on ice for 3 min for permeabilization. After incubation, samples were washed in 1 mL RSB containing 0.1% Tween-20 and pelleted at $500 \times g$ for 10 min at 4 °C. Samples were then resuspended on ice in 50 µL transposition reaction mix containing 2.5 µL Tn5 transposase, 1x TD buffer (both Illumina FC-121-1030), 1x PBS, 0.1% Tween-20 and 0.01% digitonin, and incubated at 37 °C for 30 min with agitation. Tagmented DNA was purified using Zymo DNA Clean and Concentrator-5 kit (Zymo D4014). The resulting DNA was amplified for 12–13 cycles. The libraries were purified with AMPure XP beads (Beckman A63880), quality checked on Bioanalyzer (Agilent), and 75 bp paired-end sequenced on Illumina HiSeq 4000 at Genomic Core Facility at University of Bergen.

**Cut&Run and library construction**. Cut&Run was performed as described with minor modifications[40]. Briefly, $5 \times 10^5$ cells were washed and bound to concanavalin A-coated magnetic beads (Bangs Laboratories, BP531). The cells were then permeabilized with Wash Buffer (20 mM HEPES pH 7.5, 150 mM NaCl, 0.5 mM spermidine and 1x Roche Complete Protease Inhibitor, EDTA-free) containing 0.025% digitonin (Digitonin Buffer) and 2 mM EDTA and incubated with primary antibody (anti-C/EBPB or IgG isotype control) overnight at 4 °C. The cell-bead slurry was washed twice with Digitonin Buffer and incubated with 1x Protein-A/G-MNase (pAG-MNase; Epicypher) in Digitonin Buffer for 10 min at room temperature. The slurry was washed twice with Digitonin Buffer and incubated in Digitonin Buffer containing 2 mM CaCl₂ for 2 h at 4 °C to activate pAG-MNase digestion. The digestion was stopped by addition of 2x Stop Buffer (340 mM NaCl, 20 mM EDTA, 4 mM EGTA, 50 µg/mL RNase A, 50 µg/mL GlycoBlue and 300 pg/mL in-house MNase-digested yeast spike-in chromatin) and the sample was incubated for 10 min at 37 °C to release chromatin to the supernatant and degrade RNA. The supernatant was recovered, and DNA was isolated through phenol-chloroform extraction and ethanol precipitation. Libraries were constructed to enrich for sub-nucleosomal fragments using the NEBNext® Ultra™ II DNA Library Prep Kit for Illumina as described (NEB, E7645S). The libraries were size-selected and purified with AMPure XP beads, quality checked on Tapestation (Agilent), and 100 bp or 75 bp paired-end sequenced on MiSeq or HiSeq 4000 at Genomic Core Facility at University of Bergen.

**ClonTracer barcoding of cancer cell lines and in vivo implantation**. The ClonTracer barcoding library was obtained from Addgene (#67267). The library was electrotransformed as described[25], expanded, extracted, and pooled together. For viral production, HEK293T cells were transfected with 12 µg Gag/Pol plasmid, 6 µg envelope plasmid, and 12 µg ClonTracer library. Six million E0771 cells were infected by lentiviral ClonTracer barcodes at a multiplicity of infection (MOI) of around 0.1 and infected cells were selected with puromycin. After selection, infected cells were pooled and expended in vitro. At the day of injection, cells were harvested and counted. 10,000 cells were suspended with 50% (v/v) matrigel and injected into each chow or HFD-fed mice and three replicates were set up for each condition. And five million cells were washed with PBS and the cell pellet was

stored at −80 °C as a pre-injection control for further process. At 18 days post-injection, the mice were sacrificed and the snap-frozen tumors were stored at −80 °C for further process.

**Barcode amplification and sequencing.** For each sample, the frozen tumor was crushed and the tumor pieces from different areas (tumor core, intermediate and peripheral layers) were weighed for DNA extraction. Genomic DNA was extracted from around 130 mg frozen tumor tissues cell with a Tissue DNA Kit (E.Z.N.A). PCR was used to amplify the barcode sequence for NGS, and PCR primer sequences information is listed in the Supplementary Information (Supplementary Table 3). For each PCR reaction, 2 μg of genomic DNA was used as a template and eight parallel PCR reactions were set up to ensure the sampling of sufficient template coverage. PCR products were cleaned up by PCR Purification Kit (QIAGEN) and further purified with AMPure XP beads (Beckman A63880). After purification and quality control, the samples were 75 bp paired-end sequenced on MiSeq Genomic Core Facility at University of Bergen.

**Mass cytometry.** Cells were plated in 10-cm plates in triplicates to reach a confluency of 80% after 48 h. For the analysis, cells were collected using TrypLE Express (Gibco, 12604-021). $1 \times 10^6$ cells per condition were included. Cells were resuspended in cell culture media and treated with 0.25 μM Cisplatin for 5 min at RT. Further, cells were fixed in 1 mL of 1.6% PFA (Electron Microscopy Sciences, 15710) in PBS for 10 min at RT. Cells were pelleted by centrifugation for 5 min at $900 \times g$ and the pellets were stored at −80 °C until staining with heavy metal tagged antibodies. On the day of staining, samples were thawed on ice, resuspended in 500 μL of DPBS (Gibco, 14040-133), and incubated for 10 min at RT in DPBS/DNases (Sigma, DN25) solution. Next, cells were washed in D-WASH solution (DPBS + 1% FA-free BSA + 0.02% NaN₃ + 0.25 mg/mL DNase) and barcoded (Fluidigm, 201060) according to the manufacturer's protocol. Cells were then washed twice in the Maxpar Cell Staining Buffer (Fluidigm, 201068), all samples were combined and labeled with surface antibody cocktail (Fig. S2A, extracellular) for 30 min at RT. Further, cells were pelleted by centrifugation and incubated in 4 mL of DPBS/DNase solution for 10 min at RT. After this step cells were washed in PBS-EDTA and fixed in 2% PFA/PBS (filtered through a 0.22 μm filter) for 30 min RT, followed by wash in Cell Staining Buffer and permeabilization in cold methanol (−20 °C) for 10 min. After incubation, cells were washed in once PBS, once in D-WASH, and labeled with intracellular antibody cocktail (Supplementary Table 2, intracellular) for 30 min at RT. This was followed by incubation of cells in D-WASH for 10 min at RT and double wash in D-WASH. Then cells were incubated in 2% PFA/PBS with Cell-ID™ Intercalator-Ir (Fludigm, 201192B) at 4 °C overnight. The samples were spun down the following day and incubated in D-WASH for 10 min at RT, washed once in PBS/EDTA, 3 times in Maxpar Water (Fluidigm, 201069), resuspended in EQ Four Element Calibration Beads (Fluidigm, 201078) diluted 1:9 in water, and acquired on the Helios - Mass Cytometer.

**Mass cytometry data analysis.** FCS-files were normalized, concatenated, and debarcoded in R using CATALYST[75]. Samples from the different conditions were subsampled (85,000 for HCC1806 cell lines and 100,000 per sample MDA-MB-231 cell lines) prior to analysis. Dimensionality reduction with tSNE, density plotting, and pseudo coloring were performed in Cytobank[76].

**Survival analysis.** Patients were stratified into two groups by BMI 25. Disease-Specific survival (DSS) Kaplan–Meier curves were generated using GraphPad Prism software and statistical significance was calculated using Log-rank (Mantel–Cox) test.

**Mutual information.** Mutual information was calculated as described in Goodarzi 2009[35]. Briefly, ~300 genes were used to perform gene-set enrichment analysis using iPAGE. The gene expression changes in PA-adapted cells relative to their parental line were analyzed and visualized by volcano plot. iPAGE divided the spectrum of log-fold changes into equally populated bins and used mutual information to assess the non-random distribution of the query gene-set among these bins. We have included the mutual information value (MI) and its associated z-score reported by iPAGE. For visualization, the enrichment/depletion of the query gene-set was determined using the hyper-geometric test and the resulting p-value was used to define an enrichment score that is shown as a heatmap across the expression bins. The gene expression of parental and adapted cell lines was measured by RNA-seq. For mRNA expression of human breast cancer samples were measured by microarray analysis by using the same breast cancer patients cohort of survival analysis.

**Tissue microarray analysis.** The ten CD133-stained and ten Axl-stained TMA slides were scanned with an Aperio Scanscope CS Slide Scanner. The breast cancer cores were 1.2 mm in diameter with up to 4 cores per patient. Full analysis was performed on valid cores for patients 50 years and older with ER and PR negative status. Cores with too few cells, poor quality, excessive tearing, or folding were not considered valid and were omitted from analysis.

QuPath (version 0.2.0-m5) was used to dearray the TMAs, segment cells, and classify cell types. The following detection steps and parameters were applied to all TMA slides. Simple tissue detection was used to find the approximate tissue borders within each dearrayed TMA core. For CD133-stained cores, a threshold of 229 (default 127), requested pixel size of 1 μm (default 20 μm), and checking the box for Expand boundaries were found to be the most important parameter setting changes for accurate tissue detection.

Watershed cell detection was used to create cell masks within the detected tissue of each valid core. The watershed parameters were optimized to detect large weakly hematoxylin-stained cancer cells, to minimize false-positive cell detection from areas of high background signal, and to reduce the creation of cell masks that spanned multiple cells. The watershed parameter changes deemed most important for accurate cell mask creation were: nucleus background radius of 10 μm (default 8 μm), nucleus minimum area of 24 μm² (default 10 μm²), nucleus maximum area of 230 μm² (default 400 μm²), intensity parameters for threshold and max background both set to 0.07, and exclusion of DAB staining (as was recommended for membrane staining markers). In addition, the cell expansion was set to 10 μm, 5 μm larger than the default setting, in order to capture the CD133 membrane staining on the large cancer cells.

Annotation objects were drawn around easily defined areas that contained primarily cancer cells, non-cancer cells, or platelets/RBCs and labeled as the classes tumor, stroma, or ignore, respectively. Platelets/RBCs were ignored because they appeared brown even before staining and show up as falsely positive for CD133. 9039 cells from the annotation objects drawn across 5 of the 10 slides were used to train the random forest (trees) classifier in QuPath. DAB-specific measurements were excluded from the classifier-selected features. The intensity feature used to identify CD133 positive cells was Cell: DAB OD max at a threshold of 0.45. With these parameters, the detection classifier created seven classification groups of cells: total (base) tumor cells, total stroma cells, CD133⁺ tumor cells, CD133⁻ tumor cells, CD133⁺ stroma cells, CD133⁻ stroma cells, and ignored cells. Cell masks from cores with partial low quality due to folding or poor imaging were removed to prevent false-positive cells. All cores were visually inspected for false-positive cancer cell masks and false-positive masks were removed. Mean CD133⁺ cancer cell percentage was calculated for each patient for all valid tumor cores by QuPath and exported to MS Excel. Patients with greater than 2% CD133 positive cancer cells were considered to have CD133 positive tumors.

For Axl signal analysis, a few parameters were modified from CD133-staining analysis due to the difference of staining pattern. A threshold of 228 was used for simple tissue detection. The modified watershed parameters were nucleus maximum area of 400 μm², max background intensity set to 2, and no exclusion of DAB. The total number of training objects used to train the random forest classifier was 7230 from across all the 10 slides. The ninetieth percentile of the Cell: DAB OD max intensity feature for all cells in 8 of 10 TMAs were used as a threshold to identify Axl^high cells. This resulted in Cell: DAB OD max threshold set to 1.46492. With these parameters, we identified the detection classifier created seven classification groups of cells: total (base) tumor cells, total stromal cells, Axl^high tumor cells, Axl^(−/low) tumor cells, Axl^high stroma cells, Axl^(−/low) stroma cells, and ignored cells. The number of detected cells in each cell group for each patient core were exported from QuPath. In MS Excel, this data was used to find the percentage of Axl^high cells of all tumor cells present for each ER⁻PR⁻ postmenopausal patient. An outlier value deviating more than 2 times the standard deviation in the Axl^high BMI ≤ 25 group was excluded prior to statistics performed. Statistical analysis was performed in GraphPad Prism v8.4.1.

**Student's t-test.** Statistical analysis of flow cytometry data was performed using Student's t-test on GraphPad Prism v8.4.1 software.

**Limiting dilution analysis.** The frequency of tumor-initiating cells was calculated using the Extreme Limiting Dilution Analysis (ELDA) (http://bioinf.wehi.edu.au/software/elda/index.html)[77].

**RNA sequencing data analysis.** Sequenced reads were quality checked with FastQC and aligned to the UCSC hg19 reference genome with Hisat2. Aligned reads were counted and summarized for the annotated genes using featureCounts. Differential gene expression analysis was performed by DESeq2. For visualization, read counts were normalized and regularized log transformed (rlog) for cross-sample comparison or were converted to fragments per kilobase of transcript per million mapped reads (FPKM) for within-sample comparison.

**ATACseq data analysis.** ATACseq reads were quality checked with FastQC[78] before and after adapter trimming with Trimmomatic[79]. The trimmed reads were aligned to the UCSC hg19 or mm10 reference genome using Bowtie2[80] with the parameters --phred33 --end-to-end --very-sensitive -X 2000. Reads were then removed if they were mapped to the mitochondria and non-assembled contigs, had a mapping quality score below 10, and were PCR duplicates. Read start sites were adjusted for Tn5 insertion by offsetting +stand by +4 bp and −strand by −5 bp as previously described[76]. For peak calling, MACS2[81] was used with the parameters -q 0.01 --nomodel. Peaks residing in the ENCODE blacklisted regions were removed for further downstream analysis. deepTools[82] was used to generate 1x normalized

bigwig files for visualization. ATACseq libraries were quality-controlled according to ENCODE standards.

Peaks unique to or shared across conditions were identified using the occupancy mode in DiffBind[83]. Subsequently, differential analysis was performed on these peaks with default settings using a false discovery rate <0.05, and annotated genome-wide with respect to the closest transcription start site with ChIPseeker[84]. To infer differential transcription factor binding motif activity between conditions, diffTF[37] was used. Significantly differential transcription binding motif activity was defined using a false discovery rate <0.05. Input transcription factor binding sites for 640 human transcription factors were generated as described using the HOCOMOCO database and PWMscan (cutoff $p$-value—0.00001, background base composition—0.29;0.21;0.21;0.29). HOMER motif enrichment analysis using the same HOCOMOCO motifs was included as a comparison.

To infer the differentiation state, chromVar was applied to the consensus peakset in E0771 ex vivo cells and extended to published data[85] on mammary gland development to generate a matrix of average transcription factor binding motif activity of each cell type. The data matrix was then standardized and scaled by calculating the z-scores of each transcription factor motif activity across the samples and used for principal component analysis (PCA). The difference from the fetal mammary stem cell stage was quantified by the differences in z-scores between E0771 HFD and Chow from the mean PC1 and PC2 of fetal mammary stem cell stage. To visualize the motif activity of specific transcription factors along the mammary gland developmental trajectory, snATACseq data on mammary gland development was analyzed as originally described[86] and presented in pseudotime.

**Cut&Run data analysis.** Cut&Run reads were quality checked with FastQC before and after adapter trimming with Trimmomatic. The trimmed reads were separately aligned to the UCSC hg19 and sacCer3 reference genomes using Bowtie2 with the parameters --local --very-sensitive-local --no-unal --no-mixed --no-discordant --phred33 -I 10 -X 700 and --local --very-sensitive-local --no-unal --no-mixed --no-discordant --phred33 -I 10 -X 700 --no-overlap --no-dovetail, respectively. Reads were then removed if they were mapped to the mitochondria and non-assembled contigs and had a mapping quality score below 10. Mapped reads were converted to paired-end BED files containing coordinates for the termini of each read pair and the fragment length, and calibrated to the yeast spike-in using spike_in_calibration.csh (https://github.com/Henikoff/Cut-and-Run/) in bedgraph formats for visualization. Peaks were called with SEACR[87] with respect to the IgG control using the non and stringent mode. Peaks overlapping with the ENCODE blacklisted regions were removed for further downstream analysis.

To identify enriched motif sequences protected by transcription factor binding independent of the peak-calling algorithm, pA/G-MNase cutting footprints were detected. Ends of all CUT&RUN fragments ≤120 bp were enumerated to determine the precise single base pair cut sites and sorted. Footprints were detected using Footprint Occupancy Score (FOS)[43]. Significant footprints with FOS ≤ 1 were analyzed for enriched motif sequences with HOMER using the position weight matrices (PWMs) from the HOCOMOCO database[88].

Peaks unique to or shared across conditions were identified using the occupancy mode in DiffBind to generate a consensus peakset. Raw counts of this peakset across samples were input to DESeq2 with the inverse of the spike-in calibration factors as sizeFactors to perform differential analysis. Differential peaks were annotated with respect to the closest transcription start site with ChIPseeker. HOMER was used for de novo motif analysis on differential peaks unique to each condition, and the identified motifs were compared to the HOCOMOCO database for best matches.

**ClonTracer barcode analysis.** Barcode-composition analysis was carried out by using the python package clonTracer v1.2[25]. As previously described, only barcodes passing all the quality filters and seen at least twice are considered for the analysis. To estimate the barcode distribution in each group, we first excluded the unique barcodes which were not presented in the pre-injection control cell sample from tumor samples and further a threshold of minimum ten reads per unique barcode was set up to identify the barcodes in each tumor. After calculating the fractions of barcode in each tumor sample, we pooled the data for all tumors in each group. The overall distributions of relative barcode size for chow and HFD groups were plotted in GraphPad Prism.

**Reporting summary.** Further information on research design is available in the Nature Research Reporting Summary linked to this article.

## Data availability

All data generated and analyzed during this study are available within the Article and Supplementary Files, or available from the authors upon request. The sequencing data (RNA-seq, ATACseq, Cut&Run, and ClonTracer barcode assay) generated in this study have been deposited in the European Nucleotide Archive (ENA) at EMBL-EBI under accession number PRJEB 39793 and are available at the following link. The mass cytometry data has been deposited in the FLOW Repository under repository ID FR-FCM-Z2TK and are available at the following link: https://flowrepository.org/id/FR-FCM-Z2TK. The raw microarray data are not publicly available due to national

regulations regarding patient privacy. The data may be available upon reasonable request and pending project-specific ethics approval. Requests may be directed to Stian Knappskog (email: stian.knappskog@uib.no). Source data are provided with this paper.

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

## Acknowledgements

We thank Erik Løkkevik, Bjørn Østenstad, Steinar Lundgren, Terje Risberg, and Ingvil Mjaaland for providing clinical samples. We thank the genomic score facility (GSF) at the University of Bergen, which is a part of the NorSeq consortium, provided services on RNA-seq, ATACseq and Cut&Run. GSF is supported by grants from the Research Council of Norway (245979/F50) and the Trond Mohn Foundation (BFS2016-genom). The flow cytometry and mass cytometry were performed at the Flow Cytometry Core Facility, Department of Clinical Science, University of Bergen. Helios Mass Cytometer was supported by the Trond Mohn Foundation. We thank Ingeborg Winge from the Department of Pathology, Haukeland University Hospital, Bergen, for provided training and help with TMA Immunohistochemistry. Hani Goodarzi is supported by R01CA240984 and R01GM123977. N.H. was funded by a Starter Grant from the Trond Mohn Foundation, the Norwegian Research Council (275250), and the Norwegian Cancer Society (212734-2019; National Group of Expertise on Pancreatic Cancer).

## Author contributions

Conceptualization: N.H.; methodology: N.H., X.L., A.R., S.M.G., M.H.C., S.T.T., and L.P.; software, S.M.G., C.E.W., X.L., and M.H.C. validation: T.L., A.R., and M.H.C. Formal analysis: C.E.W., N.M., P.E.L., S.K., H.G., S.D.P., S. T.T., X.L., A.R., and M.H.C.; investigation: A.R., X.L., and M.H.C.; resources: N.H., S.D.P., S.M.G., J.L., S.K., P.E.L., S.K., and A.M.; writing—original draft: N.H., X.L., and A.R.; visualization: N.H., A.R., X.L., M.H.C., and C.E.W.; supervision: N.H.; funding acquisition: N.H.

## Competing interests

The authors declare no competing interests.

## Additional information

**Peer review information** *Nature Communications* thanks Chunliang Li, Max Wicha and the other anonymous reviewer(s) for their contribution to the peer review this work. Peer reviewer reports are available.

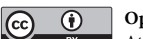

