## [Peer Review File · Nature Communications]

Reviewers' Comments:

Reviewer #1:

Remarks to the Author:

The authors have been very responsive to the revisions requested in the last round of review. They have added a large amount of new data to support their conclusions and increase the rigor of their research. On this basis, I support publication of the paper.

Reviewer #2:

Remarks to the Author:

I appreciate the additional experiments and revisions made by the authors to substantially improve this manuscript. The authors have responded to all of my previous comments and concerns.

Reviewer #3:

Remarks to the Author:

The authors have done a thorough job responding to previous critiques.

The one point that should be added is a more thorough discussion of potential reasons that this mechanism is limited to triple negative breast cancer compared to the more common Er+ subtypes.

Reviewer #4:

Remarks to the Author:

This reviewer was invited particularly to comment on the ATAC-seq related experiments. In general, the data analysis is preliminary, given the following concerns.

1. There is no detailed information associated with quality control of ATAC-seq. For instance, according to the method description, this general protocol usually generates 100k-500k ATAC-seq peaks across different mammalian cell types. Here, the 300-1000 differential peaks were identified in HFD/chow comparison groups. How many biological replicates were used? What's the stringency cutoff used in the statistical analysis to define the differential peaks?

2. At a genomic scale, where are those ATAC-seq peaks located, promoter, intergenic regions? I suggest showing a heap map with peak/tss center instead of peak summary in figure 3A.

3. How was the batch effect normalized when using the publicly available GSE dataset to compare with the newly generated ATAC-seq in this study shown in the PCA diagram in figure 3B?

4. In motif enrichment analysis (figure 3D), in addition, to show an ambiguous "TF motif activity," I recommend showing the homer or de novo motif enrichment detail, including % of typical motifs in targets and background controls along with NES and FDR. In most cases, without this detailed information, data interpretation can be affected by chromatin accessibility-associated artifacts.

5. Overall, it seems to me genome-wide chromatin accessibility change is minimal in HDF/chow. It will be critical to differentiate the ATAC-seq as a biological consequence from the direct C/EBPB binding occupancy change. Whether C/EBPE binding affinity was affected in these DE ATAC-seq peaks is the most crucial point. I understand it will be challenging if no acute protein depletion model of C/EBPB (e.g., AID, FKBP) is currently available. However, it is confusing that, as the authors have already shown in figure 5f, the cut/run data of selected marker genes are inconsistent with ATAC-seq status, which contradicts the hypothesis. The authors need to clarify the treatment/chromatin accessibility change/C/EBPB binding regulation axis in Par/Apa and HFD/chow models as well.

Reviewer #4 (Remarks to the Author):

This reviewer was invited particularly to comment on the ATAC-seq related experiments. In general, the data analysis is preliminary, given the following concerns.

1. There is no detailed information associated with quality control of ATAC-seq. For instance, according to the method description, this general protocol usually generates 100k-500k ATAC-seq peaks across different mammalian cell types. Here, the 300-1000 differential peaks were identified in HFD/chow comparison groups. How many biological replicates were used? What's the stringency cutoff used in the statistical analysis to define the differential peaks?

Authors' reply: We apologize for not detailing the quality controls for the ATACseq datasets. We used n=3 for the high fat diet condition and n=4 for the chow condition. For the *in vitro* selection system, we used n=3 for both parental and adapted cells. In **Supplemental Figure 3A and K** we specify the number of reads per sample after filtering, the %mitochondrial reads mapped to the initial pre-processed library, the transcriptional start site enrichment and the fraction of reads in peaks (encode standard requires >0.3; <https://www.encodeproject.org/atac-seq/>), the latter two of which indicate a high signal-to-noise ratio of our ATACseq libraries. **Supplemental Figure 3B and L** displays the fragment length distribution of a representative ATACseq reported in each study. It clearly shows a nucleosomal laddering pattern typical of an ATACseq library, with reads corresponding to the nucleosome-free regions (< ~150bp), the mono-nucleosomes (peaking around 200bp), and the di-nucleosome (peaking around 350bp). To identify significantly differential peaks, we analyzed the ATACseq data by diffBind and used a false discovery rate (FDR) <0.05. We have updated the methods section to include this.

2. At a genomic scale, where are those ATAC-seq peaks located, promoter, intergenic regions? I suggest showing a heap map with peak/tss center instead of peak summary in figure 3A.

Authors' reply: We thank the reviewer for this suggestion. We added the requested information in **Supplemental figure 3C, D and E**.

3. How was the batch effect normalized when using the publicly available GSE dataset to compare with the newly generated ATAC-seq in this study shown in the PCA diagram in figure 3B?

Authors' reply: This is a great question. We did take potential batch effects into consideration during the analysis. We used chromVar to generate a matrix of average transcription factor binding motif activity of each cell type normalized to sequencing depth. This matrix of all samples was further normalized by standardization and scaling using z-scores of each included motif in each sample of both the GSE and our own datasets before performing principal component analysis. We have now provided visualization of the first three principal components using biplots. These capture around 98% of the variance of the data (**Supplemental Figure 3F**). As shown in **Figure 3C**, the samples of the GSE dataset do not cluster to themselves or separate from the data points of our own data according to any of the first three principal components (**Supplementary Figure 3G and H**), which could have otherwise indicated a significant batch effect. This suggests that the batch effect has been corrected or was minimal in the final analysis without the need for further batch correction. We have updated the methods section to clarify this.

4. In motif enrichment analysis (figure 3D), in addition, to show an ambiguous "TF motif activity," I recommend showing the homer or de novo motif enrichment detail, including % of typical motifs in targets and background controls along with NES and FDR. In most cases, without this detailed information, data interpretation can be affected by chromatin accessibility-associated artifacts.

Authors' reply: Initially, we used diffTF because this newly reported tool is built specifically for quantitative measurement of different transcription factor motif activity based on genome-wide accessibility ATACseq data. This tool is internally built upon DiffBind. It allows for direct

quantitative measurement of differential motif enrichment of each motif (defined as motif activity) based on the log2 fold change in accessibility centered at each motif binding site while accounting for the GC content rather than enrichment of motif sequences alone in a set of user-defined differential peaks as input to tools such as HOMER. This tool has been benchmarked in comparison with other motif enrichment analysis tool, such as HOMER, in the paper that described the tool (Berest et al, Cell Reports 2019) and was found to be more sensitive. For comparison, we have now included HOMER motif enrichment analysis (**Supplementary Figure 3N and O**) of the same datasets reported in **Figure 3D**. This yielded the same result as diffTF analysis, i.e. only the C/EBPB motif was significantly enriched in both in the ATACseq peaks upregulated (unique gain peaks) in E0771 HFD and MDAapa relative to E0771 Chow and MDApar, respectively.

5. Overall, it seems to me genome-wide chromatin accessibility change is minimal in HDF/chow. It will be critical to differentiate the ATAC-seq as a biological consequence from the direct C/EBPB binding occupancy change. Whether C/EBPE binding affinity was affected in these DE ATAC-seq peaks is the most crucial point. I understand it will be challenging if no acute protein depletion model of C/EBPB (e.g., AID, FKBP) is currently available. However, it is confusing that, as the authors have already shown in figure 5f, the cut/run data of selected marker genes are inconsistent with ATAC-seq status, which contradicts the hypothesis. The authors need to clarify the treatment/chromatin accessibility change/C/EBPB binding regulation axis in Par/Apa and HFD/chow models as well.

Authors' reply: The reviewer suggests further clarification of our proposed model that links obesity-dependent chromatin accessibility changes to enhanced C/EBPB occupancy of target promoter sites and thereby increase the cancer initiation capacity. The reviewer raises four specific points:

Firstly, the reviewer comment that the genome-wide effects on chromatin accessibility is minimally affected by the obese environment. We agree that the effects observed (**Figure 3A, B**) were not as massive as seen in other systems as for example induced pluripotency. However, each experimental model is different, and the fact that merely exposing already highly malignant cancer cells (E0771) to the obese environment results in significant and specific epigenetic remodeling is both novel and highly interesting. Further, our downstream *in vitro* and *in vivo* functional analysis demonstrate that that such epigenetic activation of the C/EBPB regulon is required for obesity-induced stemness.

Secondly, the reviewer underscores the importance of quantifying C/EBPB binding in the chromatin regions opened by adaptation to palmitic acid and obesity. We fully agree and this was indeed our main motivation for performing the C/EBPB CUT&RUN experiments. In **Figure 5C** we demonstrate that specific C/EBPB binding (derived from the CUT&RUN) is highly enriched in chromatin regions made more accessible by adaptation to palmitic acid. One such region, the promoter region of LCN2 is exemplified in **Figure 5D**. The functional importance of increased C/EBPB occupancy in obese states was further established through ectopic overexpression of C/EBPB (LAP isoform). In these studies, we show that overexpression of C/EBPB only induces stemness after the cells have been adapted to the palmitic-rich environment and thereby display increased accessibility to its C/EBPB target sites (**Figure 4H, I**). When performing the same experiment in cells without adaptation and therefor closed chromatin at the C/EBPB binding site, ectopic overexpression failed to induce stem cell features (**Figures 4L, M**)

Thirdly, the reviewer highlights the need for functional protein depletion. Again, we absolutely agree. While we do not have acute depletion system as *AID* or *FKBP* available, we have performed our functional depletion studies using independent lentiviral delivered shRNAs. For these studies we have taken great care to perform the given experiment at as early passages as possible to avoid/limit potential adaptive mechanisms. With these C/EBPB depleted cells we show that C/EBPB is required for obesity-induced stemness *in vivo* and *in vitro* without

affected cellular proliferation (**Figure 4A-F**). In addition, we have complemented the depletion studies with ectopic overexpression studies as described above.

Fourthly, the reviewer suggests that the ATACseq and CUT&RUN data presented in **Figure 5F** is inconsistent with our hypothesis. We apologize, but are uncertain to what inconsistencies the reviewer refers to. As is, the data in **Figure 5F** highlights 9 genes that all display enhanced CUT&RUN and ATACseq signal intensity around the transcriptional start site accompanied by increased mRNA expression in palmitic adapted cells and obese/non-obese patients. As we show that overexpression of the LAP isoform of C/EBPB can drive the stemness phenotype in the adapted cells, we subsequently filtered this list by including expression analysis from LAP overexpressing cells (Figure 6A, B). From this we arrived at LCN2 and CLDN1 as the main downstream drivers of CEBPB-and obesity-dependent stemness *in vitro* and *in vivo*.

Taken together, the combined information gained from both the sequencing (ATACseq, RNAseq and ChIP) and the extensive functional *in vivo* and *in vitro* follow-up data in mouse and human models of obesity-induced cancer initiation, we strongly believe that we have enough evidence to substantiate our proposed model.

Reviewers' Comments:

Reviewer #4:

Remarks to the Author:

My concerns have been satisfactorily addressed.